# Cancer neoantigen prioritization through sensitive and reliable proteogenomics analysis

Bo Wen [ID] [1,2], Kai Li [ID] [1,2], Yun Zhang [ID] [1,2] & Bing Zhang [ID] [1,2][✉]

Genomics-based neoantigen discovery can be enhanced by proteomic evidence, but there remains a lack of consensus on the performance of different quality control methods for variant peptide identification in proteogenomics. We propose to use the difference between accurately predicted and observed retention times for each peptide as a metric to evaluate different quality control methods. To this end, we develop AutoRT, a deep learning algorithm with high accuracy in retention time prediction. Analysis of three cancer data sets with a total of 287 tumor samples using different quality control strategies results in substantially different numbers of identified variant peptides and putative neoantigens. Our systematic evaluation, using the proposed retention time metric, provides insights and practical guidance on the selection of quality control strategies. We implement the recommended strategy in a computational workflow named NeoFlow to support proteogenomics-based neoantigen prioritization, enabling more sensitive discovery of putative neoantigens.

---

[1] Lester and Sue Smith Breast Center, Baylor College of Medicine, Houston, TX 77030, USA. [2] Department of Molecular and Human Genetics, Baylor College of Medicine, Houston, TX 77030, USA. [✉]email: bing.zhang@bcm.edu

Proteogenomics has become a routine approach for the detection of protein sequences, resulting from genomic aberrations such as single nucleotide variants (SNVs), insertions and deletions (INDELs), RNA editing, novel junctions, gene fusions, and novel transcription regions[1–3]. This is achieved by simultaneously performing whole-exome sequencing (WES), RNA sequencing (RNA-Seq), and tandem mass spectrometry (MS/MS)-based shotgun proteomics analysis on matched samples, producing customized, sample-specific protein databases from DNA, and/or RNA sequencing data, and then searching MS/MS data against the customized protein databases. In contrast to proteomics data analysis that relies on reference protein databases alone, this approach allows the identification of peptides not included in reference protein databases, providing new opportunities to improve protein-coding genome annotation[4,5] and to identify disease-specific protein sequences[6–13].

Human tumors typically harbor multiple somatic mutations, and their translation may give rise to neoantigens, which are ideal targets for T-cell-based cancer immunotherapy because they are foreign to the immune system[14]. To identify neoantigens for personalized vaccine development, WES of matched tumor- and normal-cell DNA from individual patients has been used to identify somatic mutations, followed by prioritization of mutated alleles by RNA-Seq profiling of the tumor and subsequent in silico prediction of the binding affinity between putative neoantigens and the major histocompatibility complex (MHC)[15,16]. Because MHC binds peptides rather than RNA molecules, validation of mutated alleles through proteomic profiling will likely provide more functionally and clinically relevant neoantigens for prioritization.

Successful application of the proteogenomic approach to neoantigen prioritization relies on sensitive and reliable identification of variant peptides in proteomics data. In proteomics data analysis using reference protein databases, quality control is critical and achieved by estimating false discovery rate (FDR) using the target-decoy strategy[17]. Directly applying this strategy to proteogenomic studies without discriminating reference and variant peptides would underestimate the true FDR for variant peptides because the likelihood of experimentally observing a variant peptide is lower than that of a reference peptide. Thus, this global FDR method is prone to false-positive variant peptide identifications[3]. To address this issue, two alternative FDR estimation methods have been proposed. The separate FDR method calculates FDRs for known and variant peptides separately[18,19]. The two-stage FDR method involves two stages: in the first stage, the MS/MS data are searched against a reference protein database, and the confidently identified spectra are removed; in the second stage, the remaining spectra are searched against the variant protein database, and the FDR for variant peptides is calculated based on the second stage search results[20,21]. The separate FDR and two-stage FDR methods calculate FDRs based on a small number of identifiable variant peptides, which may generate highly variable estimates. The two-stage FDR method is further vulnerable to false negatives because an MS/MS spectrum generated from a variant peptide may be incorrectly matched to a reference peptide in the first stage and excluded from the second stage analysis. Moreover, a common limitation for all three methods is that false positives may occur when a spectrum matched to a variant peptide is actually derived from a reference peptide containing a chemical or post-translational modification that was not considered in database searching.

To strengthen quality control, multiple tools have been developed to validate the variant peptides passing the FDR threshold, such as SAVControl[22], SpectrumAI[4], and PepQuery[23]. SAVControl provides site-level quality control of single amino-acid variant (SAV) peptide identifications. It relocates the mass shift responsible for a SAV to search for alternative interpretations. SpectrumAI is specifically designed to curate SAV peptide identifications using the ions flanking both sides of the substituted amino acid. PepQuery is a peptide-centric search engine for novel peptide validation, including SAV peptides and other types of novel peptides such as those derived from INDELs. For each peptide-spectrum match (PSM), a permutation $p$ value is calculated based on randomly shuffled peptide sequences to evaluate the statistical significance of the PSM score. The peptide-centric analysis in PepQuery also allows comprehensive consideration of sequence modifications during the analysis. In general, all three tools can help reduce false positives, but they cannot rescue false negatives.

Applying different quality control strategies to the same proteomics data may result in different numbers of identified variant peptides, obscuring neoantigen prioritization and other downstream proteogenomic applications. Recent studies have compared different FDR estimation methods for novel peptide identification[21,24,25]. Li et al.[24] and Ivanov et al.[25] utilized spike-in data sets in which spiked-in proteins were considered as novel proteins. Sequences of spiked-in proteins are usually very different from those in the reference database. In contrast, most variant proteins identified in proteogenomic studies only involve very minor sequence changes compared with the reference sequences, such as SAVs. These minor sequence changes cannot be well-represented by spiked-in proteins. Moreover, existing studies have only evaluated different FDR estimation methods in label-free experiments, but many cancer proteogenomic studies involve isobaric labeling[7,10,12]. The conclusions from the label-free studies may not be generalizable to isobaric labeling-based experiments because of sample multiplexing. In addition, variant peptide validation tools introduced more recently were not included in these evaluation studies. Thus, despite the wide application of these approaches, there remains a lack of evidence-based consensus on the best bioinformatics strategy for controlling the quality of variant peptide identifications. A key challenge is the lack of an appropriate metric for systematic and unbiased performance evaluation.

Peptide retention time (RT) in a liquid chromatography tandem mass spectrometry (LC-MS/MS) experiment, i.e., the time points when peptides elute from the LC column as recorded by the instrument, is an intrinsic feature of a peptide. Recent studies have demonstrated the potential of predicting peptide RT on the basis of peptide sequence[26–31]. If the RT can be predicted with high accuracy for all peptides, the predicted RT of the peptide in each PSM can be compared with the observed RT associated with the spectrum to determine the quality of the PSMs. Although several studies have shown the value of integrating RT information into the proteomics data analysis workflow[26,32–35], RT is typically not used in peptide identification and is independent of the FDR estimation. Therefore, the difference between predicted and observed RTs can serve as an effective and unbiased quantitative metric for evaluating the quality of PSMs reported by different variant peptide identification methods.

In this study, we leverage the power of automated deep learning and transfer learning to develop AutoRT, a peptide sequence-based RT prediction tool that can predict RT with high accuracy. We use the predicted RT to systematically compare different quality control strategies for variant peptide identification in three large-scale experimental data sets generated on label-free, tandem mass tag (TMT), and isobaric tags for relative and absolute quantification (iTRAQ) platforms. Our systematic evaluation based on the analysis of 57 million spectra generated on three popular MS platforms using three search engines provides insights and practical information to guide method selection for future proteogenomic studies. We further implement the

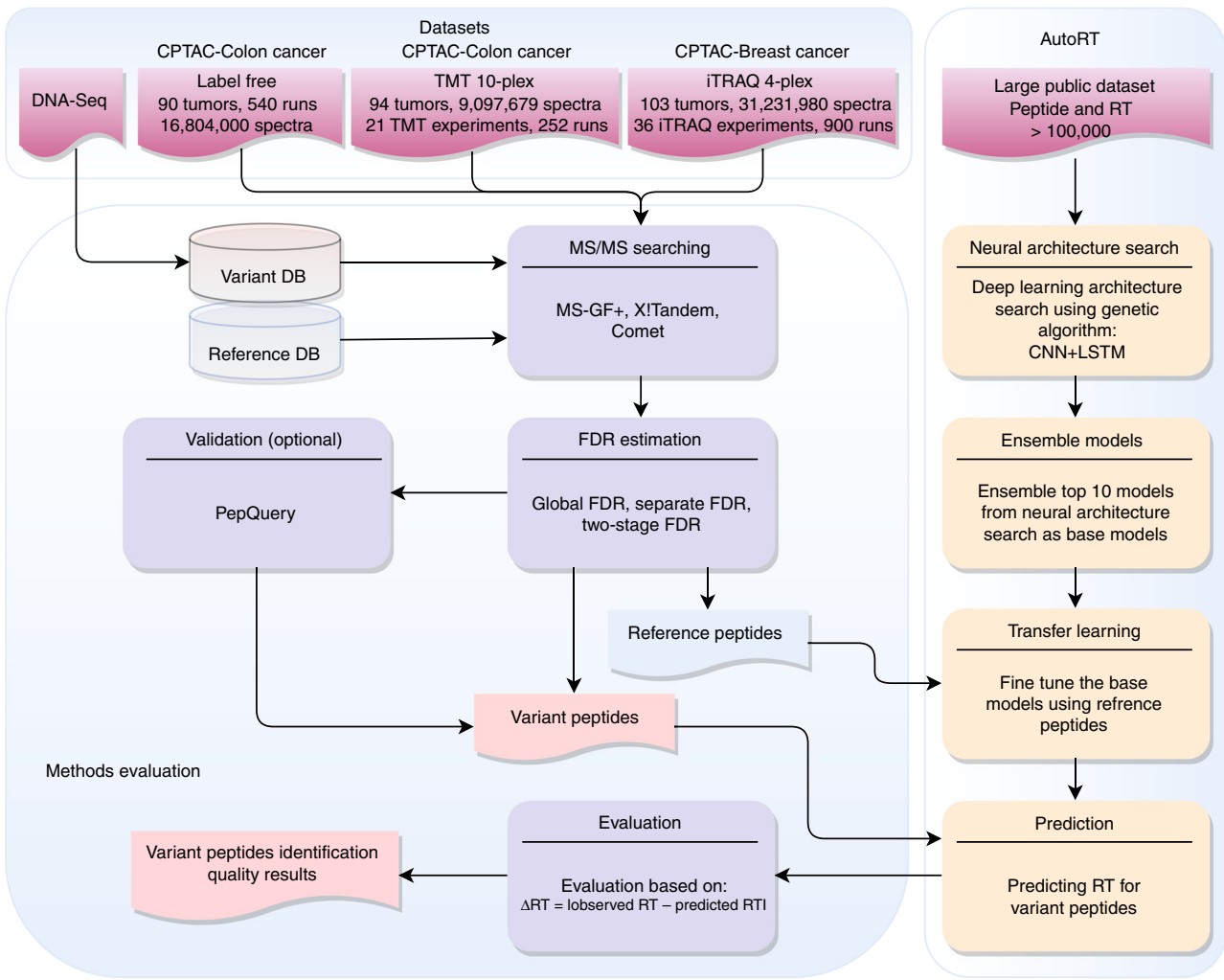

**Fig. 1 An overview of the study design.** The left panel shows the overall study design including data sets, customized database construction, search engines used, different quality control strategies, and the variant peptide identification evaluation method. The right panel shows the development and application of the peptide retention time (RT) prediction models.

recommended bioinformatics strategy into NeoFlow, a streamlined computational workflow that integrates WES and MS/MS proteomics data for neoantigen prioritization to facilitate cancer immunotherapy.

## Results

**An overview of the study.** Figure 1 provides a general overview of our study design. In order to systematically evaluate quality control strategies in proteogenomics, we used three large-scale data sets from the Clinical Proteomic Tumor Analysis Consortium (CPTAC) colon cancer[10] and breast cancer studies[7], including one label-free data set, one TMT data set, and one iTRAQ data set. The label-free data set includes 90 colon tumor samples and a total of 16,804,000 MS/MS spectra. The TMT data set includes 21 TMT 10-plex experiments on 94 colon tumor samples from the same cohort as the label-free data set and a total of 9,097,679 MS/MS spectra. The iTRAQ data set includes 36 iTRAQ 4-plex experiments on 103 breast tumor samples and a total of 31,231,980 MS/MS spectra. Matched WES data of the tumor samples were used for customized database construction. For the label-free study, we built one customized database for each tumor sample. For the TMT and iTRAQ studies, we built one customized database for each TMT or iTRAQ experiment based on WES data of all individual tumor samples in the TMT

10-plex or iTRAQ 4-plex. For the purpose of method evaluation, we included both germline and somatic variants in the customized databases. We used three popular search engines, MS-GF+, X! Tandem, and Comet, for database searching, and FDR estimation was performed at the peptide level using three FDR estimation methods: global FDR, separate FDR, and two-stage FDR, all with a threshold of 1%. PepQuery was included as an optional step to validate the variant peptides that passed the FDR threshold. Variant peptides reported by different quality control strategies were evaluated based on the difference between observed RT and predicted RT, which was calculated using the AutoRT deep learning models as described below.

**Retention time prediction.** In order to establish a systematic and unbiased metric for evaluating the quality of the variant peptide identifications, we developed a peptide sequence-based RT prediction tool named AutoRT using automatic deep learning and transfer learning techniques. The workflow of AutoRT is shown in the right panel of Fig. 1 and described in detail in the Methods section. In brief, a large public data set containing 174,182 peptides was employed for automatic deep neural network architecture search using a genetic algorithm. Based on validation mean squared error (MSE), the top 10 best neural architectures

**Fig. 2 Peptide retention time prediction performance evaluation. a** Benchmarking AutoRT against three deep learning-based retention time (RT) prediction tools (Prosit, GuanMCP2019, and DeepMass) and one traditional machine learning-based tool GPTime based on median absolute errors (MAEs) of the predicted peptide RTs in three data sets. **b–d** MAE distribution for the CPTAC label-free **b**, TMT **c**, and iTRAQ **d** data sets, respectively. Each data point in the scatter plots represents a single LC-MS/MS run, and the numbers are the median MAEs (minute) across all runs in a data set. The boxplots on the top show the size distribution of training data, whereas the boxplots on the right show the MAE distribution. Dashed lines are fitted lines using LOESS model. Source data are provided as a Source Data file. For boxplots, centerline indicates the median, box limits indicate upper and lower quartiles, whiskers indicate the 1.5 interquartile range and points indicate outliers.

were selected, and from which 10 models were trained as the basis for transfer learning.

To evaluate the performance of AutoRT, we compared AutoRT with four recently published tools, including three deep learning-based RT prediction tools (Prosit[27], Deep-Mass[36], and GuanMCP2019[37]) and one traditional machine learning-based tool (GPTime[30]), on three large public data sets (see Methods). For all three data sets, AutoRT outperformed all other three deep learning tools in terms of median absolute error (MAE) in the independent test data, and all deep learning tools outperformed GPTime to a large extent (Fig. 2a and Supplementary Fig. 1). We also compared the performance of the ensemble model of AutoRT with that of 10 individual models. On average, the MAE of ensemble model decreased by 25, 18, and 18% compared with individual models on the three data sets, respectively (Supplementary Fig. 2).

Next, we evaluated the performance of AutoRT on the three CPTAC data sets and compared the performance of AutoRT with GPTime on these data sets. The label-free data set included 540 LC/MS-MS runs. AutoRT outperformed GPTime across all 540 runs with a wide range of training data sizes (Fig. 2b). The median MAE in the independent test data was 0.57 min for AutoRT, which was 76% lower than the median MAE of 2.42 min for GPTime. Both AutoRT and GPTime benefited from increased training data size, but the performance of AutoRT was more stable across different data sizes.

The TMT and iTRAQ data sets included 252 and 900 LC-MS/MS runs, respectively, and the performance evaluation results

were similar to those observed in the label-free data set (Fig. 2c, d). Specifically, the median MAE for AutoRT in the TMT data set was 0.68 min, which was 66% lower than that for GPTime (2.02 min). The median MAE for AutoRT in the TMT data set was 0.63 min, which was 71% lower than that for GPTime (2.18 min). AutoRT outperformed GPTime across the entire range of data sizes in both data sets. In the iTRAQ data set, GPTime performed better for the runs with smaller data sizes compared to other runs. Peptides in these runs eluted very early from the LC column and may reflect unbound material that contain unique sequence features for easy prediction.

The training sample size is relatively small in these CPTAC runs, thus, overfitting is a major concern because it may lead to poor generalization capability and significant degradation in model performance in independent testing data. To evaluate the level of overfitting, we selected three runs from each of the three data sets and compared the prediction errors on the training and independent testing data. The prediction error distributions were comparable between the training and testing data, and only a slight increase of the median was observed in the testing data (Supplementary Fig. 3). Therefore, overfitting is not a major issue here.

Together, our results demonstrate the clear superiority of deep learning over traditional machine learning for RT prediction, and AutoRT further outperformed previously published deep learning tools. The highly accurate RT predictions made by AutoRT in the label-free, TMT, and iTRAQ experiments enabled the use of predicted RT in the evaluation of the quality of variant peptide identifications.

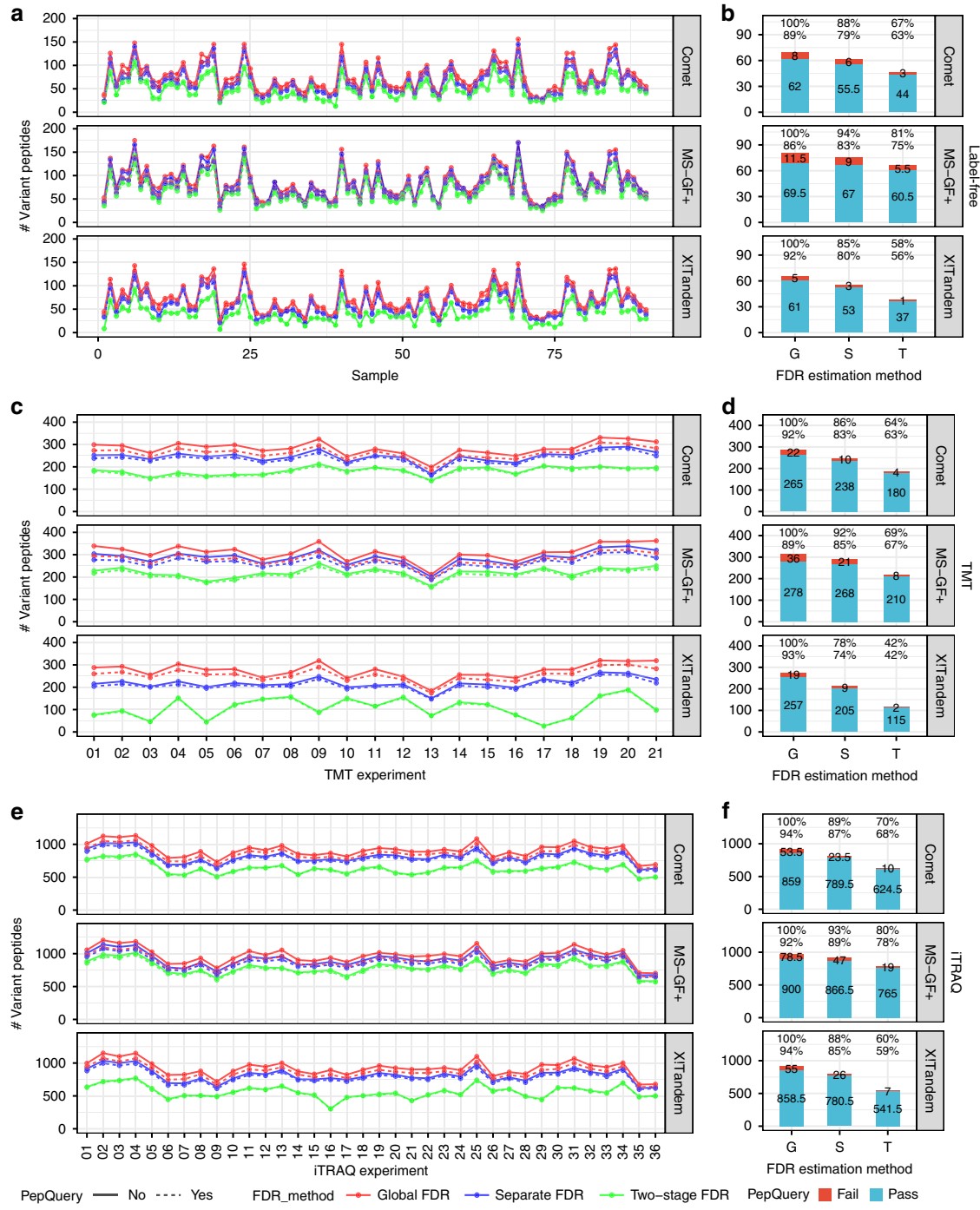

**Fig. 3 Sensitivity of variant peptide identification. a** Variant peptide identification for each label-free sample. **b** Median numbers of identified variant peptides in label-free samples. **c** Variant peptide identification for each TMT sample. **d** Median numbers of identified variant peptides in TMT samples. **e** Variant peptide identification for each iTRAQ sample. **f** Median numbers of identified variant peptides in iTRAQ samples. The percentage numbers above the bars in the bar plots are the ratios compared with the most sensitive methods in each plot. The numbers on the bars are the numbers of variant peptides. Source data are provided as a Source Data file. Solid lines indicate the identified variant peptides without filtering using PepQuery and dashed lines indicate the identified variant peptides passed PepQuery validation.

**Sensitivity of variant peptide identification**. To investigate the impact of different quality control strategies on the sensitivity of variant peptide identification, we analyzed the label-free data of the 90 tumor samples, the TMT data of the 21 TMT 10-plex experiments, and the iTRAQ data of the 36 iTRAQ 4-plex experiments. All identified variant peptides are listed in Supplementary Data 1–3.

The numbers of variant peptides identified for individual label-free samples using different search engines and quality control strategies are shown in Fig. 3a. Results from all three search engines showed similar trends across samples. On average, global FDR identified more variant peptides than the other two FDR estimation methods for all three search engines, both with and without PepQuery validation, and separate FDR identified more variant

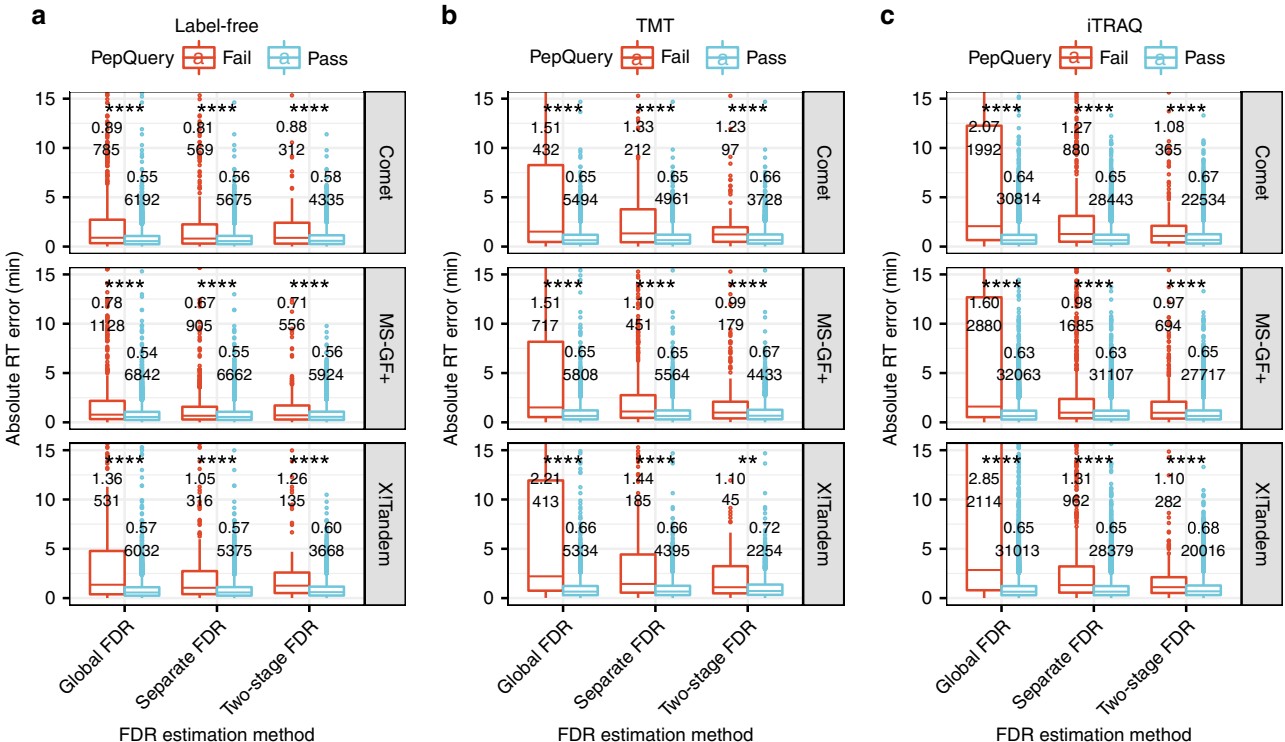

**Fig. 4 Variant peptide identification quality evaluation.** Absolute retention time (RT) error distributions for variant peptide identifications in the label-free data set **a**, the TMT data set **b** and the iTRAQ data set **c**. Numbers in the first rows of the boxplots are the median absolute RT error values. Numbers in the second rows are the number of identified variant peptides summed across all samples in each group. ns: $p > 0.05$; *: $p \leq 0.05$; **: $p \leq 0.01$; ***: $p \leq 0.001$; ****: $p \leq 0.0001$ (Wilcoxon rank sum test). Source data are provided as a Source Data file. For boxplots, centerline indicates the median, box limits indicate upper and lower quartiles, whiskers indicate the 1.5 interquartile range and points indicate outliers. The $Y$ axis was limited to up to 15.

peptides than two-stage FDR (Fig. 3b). These results are consistent with our understanding of the different levels of stringency of the three FDR estimation methods. Across the three search engines, an average of 8–14% variant peptides identified by global FDR failed PepQuery validation, and the failure rates were 5–12% for separate FDR and 3–8% for two-stage FDR. Using the numbers of variant peptides reported by global FDR without PepQuery validation, the strategy with the highest sensitivity, as 100%, the other quality control strategies reported 56–94% variant peptide identifications.

We repeated the analyses on the 21 TMT and 36 iTRAQ experiments, respectively (Fig. 3c–f). Similar to our observation in the label-free data set, global FDR identified more variant peptides than the other two FDR estimation methods for all three search engines, both with and without PepQuery validation, and separate FDR identified more variant peptides than two-stage FDR. Although the label-free and TMT experiments were performed on the same cohort of colon tumor specimens, we identified three- to fourfold more variant peptides in individual TMT experiments compared to individual label-free samples across all settings. This was expected because each TMT experiment included multiplexed samples, increasing the diversity of variants. For the TMT data, an average of 7–11% variant peptides identified by global FDR failed PepQuery validation across the three search engines, and the failure rates were 4–7% for separate FDR and 2–4% for two-stage FDR (Fig. 3d). Using the numbers of variant peptides reported by global FDR without PepQuery validation as 100%, the other methods reported 42–93% variant peptide identifications. For the iTRAQ data, an average of 6–8% variant peptides identified by global FDR failed PepQuery validation across the three search engines, and the failure rates were 3–5% for separate FDR and 1–2% for two-stage FDR (Fig. 3f). Using the numbers of variant peptides reported by

global FDR without PepQuery validation as 100%, other methods reported 59–94% variant peptide identifications. Together, these data showed that applying different quality control strategies to the same data set may lead to substantially different numbers of identified variant peptides.

**Retention time-based quality evaluation.** To evaluate the quality of the variant peptide identifications, we calculated the difference between predicted RT and observed RT for all peptides identified in each label-free sample or each multiplexed sample. Because PepQuery invalidated a subset of identified variant peptides for all FDR estimation methods and search engines considered, we first asked whether peptides that passed PepQuery validation had better quality compared with those that failed. For all three data sets, variant peptides failing PepQuery validation showed significantly higher absolute RT error compared with the ones passing PepQuery validation in all comparisons ($p < 0.01$, Wilcoxon rank sum test, Fig. 4). For variant peptides failing PepQuery validation, the ones identified by global FDR had clearly higher median absolute RT errors compared with those identified by separate FDR or two-stage FDR. In contrast, variant peptides passing PepQuery validation showed similar median absolute RT errors independent of the FDR estimation methods, and the RT errors were comparable to those for reference peptides (Fig. 2b–d). These data showed that PepQuery was highly effective in removing low quality variant peptide identifications, and that variant peptides passing PepQuery validation were of high quality for all FDR estimation methods.

Among all variant peptides passing PepQuery validation, almost all reported by the separate FDR and two-stage FDR methods were also reported by global FDR (Fig. 5a, d, g). This

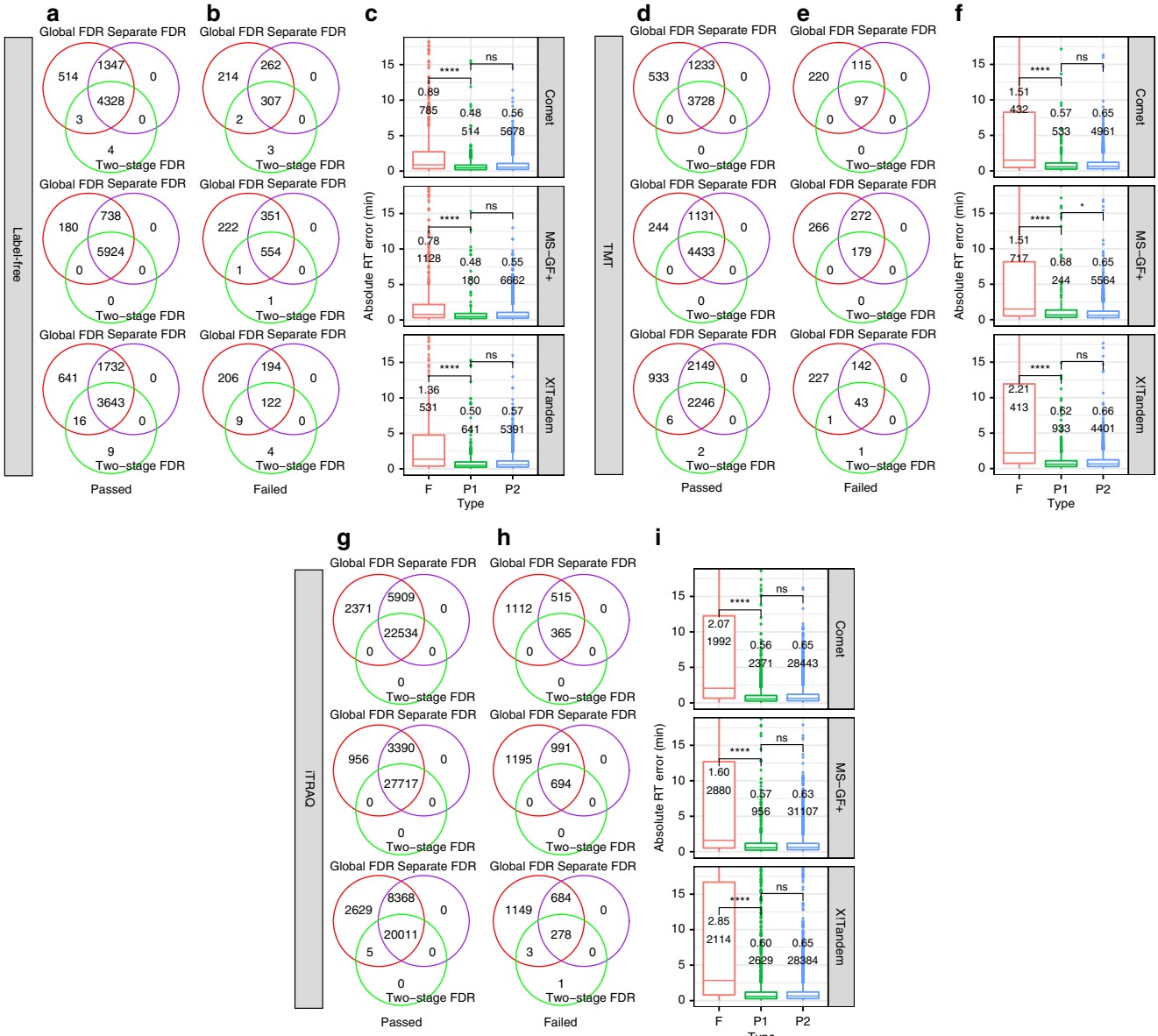

**Fig. 5 Comparison of variant peptide identifications with different levels of support. a–b** Venn diagrams of variant peptides identified by different methods that passed **a** or failed **b** PepQuery validation in the label-free data set. **c** Absolute retention time (RT) error distributions for variant peptide identifications with different levels of support in the label-free data set. **d–e** Venn diagrams of variant peptides identified by different methods that passed **d** or failed **e** PepQuery validation in the TMT data set. **f** Absolute RT error distributions for variant peptide identifications with different levels of support in the TMT data set. **g–h** Venn diagrams of variant peptides identified by different methods that passed **g** or failed **h** PepQuery validation in the iTRAQ data set. **i** Absolute RT error distributions for variant peptide identifications with different levels of support in the iTRAQ data set. F represents all variant peptides failing PepQuery validation, P1 represents PepQeury-validated variant peptides uniquely reported by global FDR, and P2 represents PepQuery-validated variant peptides reported by global FDR and at least one of the other two methods. Numbers in the first rows of the boxplots are the median absolute RT error values. Numbers in the second rows are the number of variant peptides in each group. ns: $p > 0.05$; *: $p \leq 0.05$; **: $p \leq 0.01$; ***: $p \leq 0.001$; ****: $p \leq 0.0001$ (one-sided Wilcoxon rank sum test). Source data are provided as a Source Data file. For boxplots, centerline indicates the median, box limits indicate upper and lower quartiles, whiskers indicate the 1.5 interquartile range and points indicate outliers. For boxplots, the $Y$ axis was limited to up to 18.

was also true for variant peptides that failed PepQuery validation (Fig. 5b, e, h). We further compared the quality of all variant peptides failing PepQuery validation (F), PepQeury-validated variant peptides uniquely reported by global FDR (P1), and PepQuery-validated variant peptides reported by global FDR and at least one of the other two methods (P2). As shown in Fig. 5c, f, i, P1 peptides showed significantly lower absolute RT error than F peptides in all comparisons ($p < 0.0001$, one-sided Wilcoxon rank sum test). In contrast, there was no statistically significant difference between P1 peptides and P2 peptides for eight out of the nine comparisons. These data suggest that variant peptides uniquely reported by global FDR were of high quality as long as they passed PepQuery validation. Thus, global FDR control followed by PepQuery validation offered the highest sensitivity without compromising the quality of variant peptide identifications.

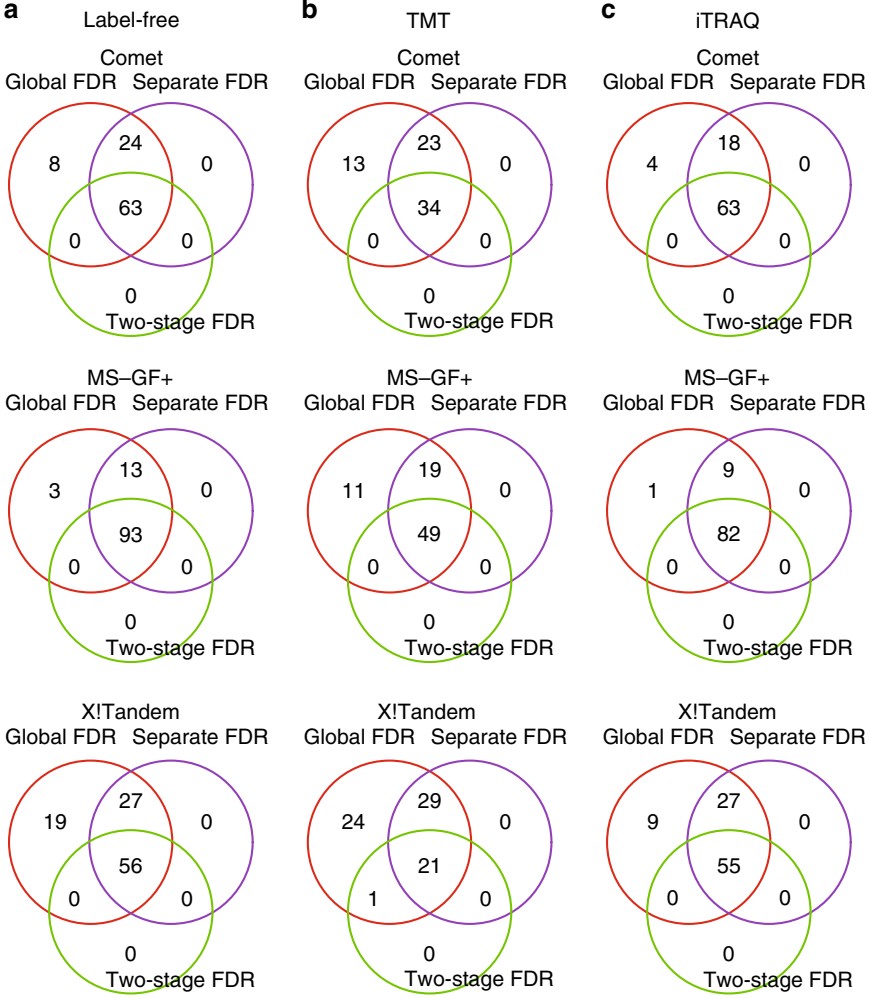

**Fig. 6 Comparison of somatic mutations supported by variant peptide identifications.** Venn diagrams of somatic mutations supported by variant peptides identified by different methods in the label-free data **a**, the TMT data **b**, and the iTRAQ data **c**. Source data are provided as a Source Data file.

In addition to PepQuery, other tools such as SAVControl and SpectrumAI have also been developed for variant peptide validation. We tried to apply these tools to validate all variant peptides passing global FDR control in the iTRAQ data set, in which the largest number of variant peptides were identified. Because the three search engines we used in this study could not generate SAVControl-compatible data formats, we were not able to apply SVAControl to our search results. SpectrumAI validated 28,852, 30,689, and 28,863 variant peptides (peptide sample pairs) for the search results from Comet, MS-GF+, and X!Tandem, respectively. These numbers were 6%, 4%, and 7% lower than those validated by PepQuery, with 96, 94, and 95% overlap. As shown in Supplementary Fig. 4, variant peptides uniquely validated by PepQuery (P1) showed similar quality compared with the ones validated by both PepQuery and SpectrumAI (P2); however, those uniquely validated by SpectrumAI (P3) showed obviously higher RT errors. These results suggest that SpectrumAI will unlikely provide significant added value beyond PepQuery for variant peptide validation.

**Neoantigen prioritization.** Our method evaluation included both germline and somatic variants, but only somatic mutations produce neoantigens. We compared the numbers of somatic mutations supported by PepQuery-validated peptides identified by different search engines with different FDR estimation methods (Fig. 6, Supplementary Data 4–6). For the label-free data set

(Fig. 6a), global FDR reported peptide evidence for 95, 109, and 102 somatic mutations using Comet, MS-GF+, and X!Tandem, respectively, and these numbers were 70, 79, and 75 for the TMT data set (Fig. 6b), and 85, 92, and 91 for the iTRAQ data set (Fig. 6c). On average, separate FDR and two-stage FDR only provided evidence for 88% and 63% of somatic mutations that were supported by global FDR.

For neoantigen prediction, we further predicted MHC-I binding affinity for all peptides of 8–11 amino acids in length that contained one of the somatic mutations supported by at least one PepQuery-validated peptide. Only mutant peptides with high predicted binding affinity ($\leq$ 150 nM) to MHC-I were considered putative neoantigens (Supplementary Data 7–9). As shown in Fig. 7a–c, different search engines and FDR estimation methods reported very different numbers of putative neoantigen-encoding mutations from the same data set. Combining global FDR results from the three search engines resulted in the identification of putative neoantigens for 27 (30%), 25 (27%), and 25 (24%) tumor samples for the label-free, TMT, and iTRAQ data sets, respectively. These numbers represent an average increase of 11% from those reported by MS-GF with global FDR control, the best single search engine setting tested, and an average increase of 151% from those reported by X!Tandem with two-stage FDR control, the worst tested setting.

Across all three data sets, putative neoantigens identified by different search engines using different FDR estimation methods

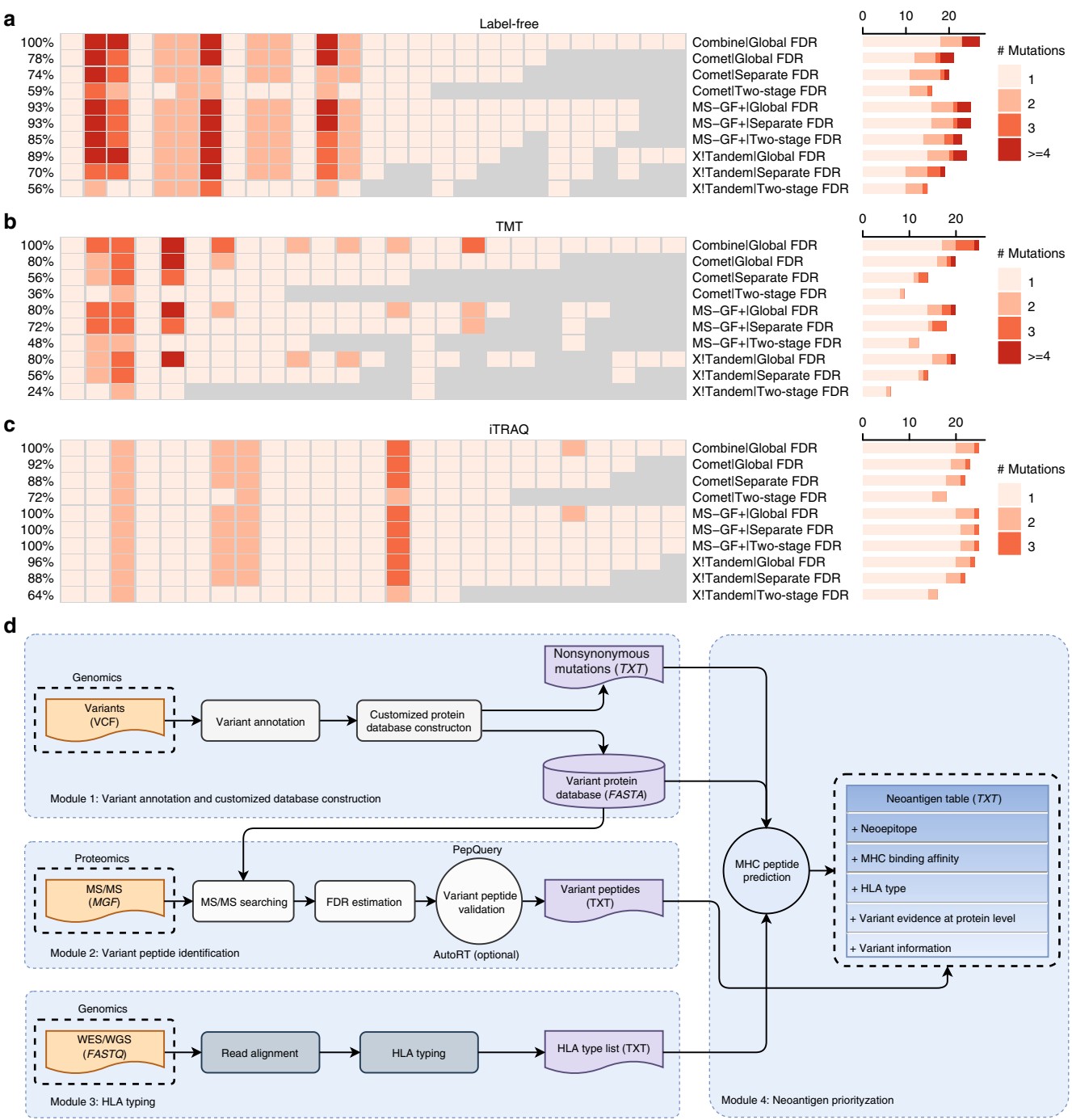

**Fig. 7 Neoantigen prediction comparison and proteogenomics workflow. a–c** Neoantigen prediction result for each sample in the label-free data **a**, the TMT data **b**, and the iTRAQ data **c**. Each column represents a sample and each row represents the neoantigen prediction result using a specific method. The color of each cell represents the number of somatic mutations with predicted neoantigens. Only putative neoantigens encoded by proteomic-supported somatic mutations were considered. **d** Overview of NeoFlow pipeline for the proteogenomics-based neoantigen prioritization. Source data are provided as a Source Data file.

in combination with PepQuery filtering showed an average median absolute RT error of 0.64 min (Supplementary Fig. 5, Supplementary Data 7–9), which was comparable to those for reference peptides (Fig. 2b, d). Despite the overall high quality of these putative neoantigen identifications, there were some clear outliers (Supplementary Fig. 5). On average, 7% of these identifications showed an RT error higher than 5 min and may thus require more critical evaluation.

To make the proteogenomics-based neoantigen prioritization method directly available to the cancer research community, we have implemented it in NeoFlow, which includes four modules as shown in Fig. 7d: (1) variant annotation and customized database construction; (2) variant peptide identification including MS/MS searching, FDR estimation, PepQuery validation, and optional RT-based validation; (3) human leukocyte antigen (HLA) typing; and (4) MHC-binding prediction and neoantigen prioritization.

These modules are described in detail in the Method section. NeoFlow is available at https://github.com/bzhanglab/neoflow.

To further demonstrate the utility of NeoFlow in the analysis of immunopeptidomics data, we applied the workflow to a published immunopeptidomics study[38]. The RT prediction models trained for this data set showed similar performance to those trained on the CPTAC data sets (Supplementary Fig. 6). NeoFlow identified nine out of the 11 somatic variant peptides reported in the original study and four additional somatic variant peptides (Supplementary Data 10). The two somatic variant peptides reported in the original paper but not identified by NeoFlow showed obviously higher absolute RT errors, suggesting the possibility of false positives. Among the four newly identified somatic variant peptides, two have been reported in a recently published reanalysis of the same data set[39]. These results demonstrate the sensitivity and specificity of NeoFlow in analyzing immunopeptidomics data and the value of RT-based validation as an additional filter to reduce false positives.

## Discussion

Applying three search engines to three cancer data sets generated on three popular MS platforms and a total of 287 tumor samples, our systematic analysis showed that substantially different numbers of variant peptides and putative neoantigens are identified depending on the quality control strategy used. Using the difference between accurately predicted peptide RT and observed RT as a quantitative metric, we demonstrated that among all quality control strategies investigated, global FDR control followed by PepQuery validation offered the highest sensitivity while identifying high quality variant peptides. We thus recommend this quality control strategy for variant peptide identification and neoantigen prioritization in future proteogenomic studies. There is a large variation in the numbers of putative neoantigens reported by the three search engines; combining results from multiple search engines can effectively maximize putative neoantigen identifications. Moreover, RT-based validation provides an additional filter to reduce false positives. We implemented this recommended strategy in a computational workflow to support sensitive and reliable proteogenomics-based neoantigen prioritization.

A major contribution of this study is the development of AutoRT, a deep learning-based RT prediction algorithm, because accurately predicted RT opens the door to systematic evaluation of variant peptide identifications. Training a deep learning model without using hand-crafted features typically requires a large amount of data to achieve good performance. In most deep proteomics studies, multidimensional liquid chromatographic technologies are used. Each sample is separated into multiple fractions and each fraction of peptides is injected into a mass spectrometer for data acquisition. In our study, the number of identified peptides for a single run ranged from 700 to 10,000 (Fig. 2b–d), which in general is not large enough to train a peptide sequence-based high-performance deep learning model from scratch. Thus, we developed base models using a large public data set and then used the transfer learning strategy to address the limitation of the small size of the run-specific training data. Combining data from multiple experiment types for training may be able to create a single model that can handle experiment-specific parameters; however, this cannot replace transfer learning because nonlinear RT shift between different runs occurs frequently even when the same LC system is used for all runs in a study (Supplementary Fig. 7). Another challenge is the design of the neural network architecture because the optimal architecture is not known a priori. We addressed this challenge by using evolutionary computation to automatically search for an optimal architecture.

Consistent with previous studies[21,24,25], we found that two-stage FDR control was more stringent than separate FDR control, which was in turn more stringent than global FDR control. Interestingly, our results showed that even the most stringent two-stage FDR control still benefited from PepQuery validation. One of the most important features of PepQuery is competitive filtering based on unrestricted modification searching in which >1000 modifications are considered, which effectively addresses the common limitation of all FDR estimation methods. As expected, global FDR control benefited the most from PepQuery validation because it underestimates FDR for variant peptide identifications. Nevertheless, its combination with PepQuery validation provided the highest sensitivity without compromising reliability.

The RT errors of variant peptides that passed PepQuery validation were comparable to those of reference peptides in corresponding data sets; however, some variant peptides had high RT errors. This may be explained by different reasons such as inaccurate RT prediction and wide elution time range for some peptides, and false-variant peptide identification is also a possible explanation. Therefore, the RT errors included in the final report of NeoFlow provide orthogonal information that facilitates candidate prioritization for experimental validation. Rather than using RT errors as an optional feature to filter PSMs, a more-effective approach to improve peptide identification is to incorporate AutoRT-derived delta RT as a feature into PSM scoring in combination with other features. However, such implementation will require graphics processing units (GPUs) for PSM scoring, and thus will be more useful when GPUs are widely accessible in proteomics laboratories.

Although the primary goal of the study was to compare different quality control strategies, our results also revealed the performance difference of different search engines in proteogenomics search. Among the three search engines investigated, MS-GF+ showed the highest sensitivity in variant peptide identification both before and after PepQuery validation (Fig. 3). Meanwhile, MS-GF+ also identified higher percentages of variant peptides that failed PepQuery validation, suggesting higher risk in bringing in more false positives when used without PepQuery validation. X!Tandem showed the lowest sensitivity among the three, and we would not recommend using this search engine by itself in proteogenomics search. However, when used together with other search engines, it may still add unique variant identifications to improve the overall sensitivity (Fig. 7).

The customized database approach has been widely implemented in proteogenomic studies since its introduction in 2012[40]. However, the potential value of identified mutant peptides as cancer biomarkers is diminished by the fact that individual mutant peptides are shared by few tumors. Therefore, the clinical utility of this proteogenomic analysis remains unclear. Here, we showed a potential clinical application of this approach by identifying and selecting neoantigens for personalized vaccine development, which does not require recurrent identification of the mutant peptides. Combining the customized database approach with MHC-I binding prediction, we predicted personalized neoantigens for 30, 27, and 24% of the patients in the label-free, TMT, and iTRAQ data sets, respectively. Although proteomic sequence coverage remains limiting, proteomics data can nevertheless enhance DNA and RNA sequencing-driven target selection pipelines[15,16] and enable more-effective prioritization for patients with proteomics-supported neoantigens. The value of this approach will continue to grow as the sensitivity of MS increases.

NeoFlow can be applied to both global proteomics data from tumor tissues and immunopeptidomics data. Immunopeptidomics data provide direct evidence of both expression and presentation of somatic variant peptides, but when immunopeptidomics data is not

available, global proteomics data from tumor tissues can provide expression evidence for somatic variants. A limitation of this study is that only somatic SNVs and INDELs were considered for neoantigen identification. It has been reported that gene fusions[41] and cancer-specific intron retention events[42] are also potential sources of cancer neoantigens. Another limitation is that variant peptide validation by PepQuery in immunopeptidomics data requires an external tool to generate reference peptide candidates to constrain the search space for no-enzyme search. In the future, we will expand NeoFlow to support the identification of peptides generated by different types of mutational events and to improve PepQuery for the analysis of immunopeptidome data. These efforts will enable more-comprehensive proteogenomics-based neoantigen prioritization or detection.

## Methods

**Data sets**. Three large-scale proteomics data sets from a colon cancer study[10] and a breast cancer study[7] were downloaded from the CPTAC data portal, including a colon cancer label-free data set (https://cptac-data-portal.georgetown.edu/cptac/s/S037), a colon cancer TMT data set (https://cptac-data-portal.georgetown.edu/cptac/s/S037), and a breast cancer iTRAQ data set (https://cptac-data-portal.georgetown.edu/cptac/s/S015). Samples for both colon cancer data sets came from a common patient cohort. The label-free data set was generated on a Q-Exactive mass spectrometer at the Vanderbilt University Medical Center. It included 90 colon tumor samples, 540 LC-MS/MS runs, and a total of 16,804,000 MS/MS spectra. The TMT data set was generated on a Q-Exactive Plus mass spectrometer at the Pacific Northwest National Laboratory. It included 94 tumor samples, 21 TMT 10-plex samples, 252 LC-MS/MS runs, and a total of 9,097,679 MS/MS spectra. The iTRAQ data set was generated on a Q-Exactive mass spectrometer at Broad institute. It included 103 tumor samples, 36 iTRAQ 4-plex samples, 900 LC-MS/MS runs, and a total of 31,231,980 MS/MS spectra. Somatic variants and germline variants for colon cancer samples were from the original publication[10]. Somatic variants and germline variants for breast cancer samples were downloaded from the genomic data commons (GDC, https://portal.gdc.cancer.gov/).

**Customized database construction**. Somatic and germline variants were annotated using ANNOVAR (v2017Jul17)[43] based on the Hg19 RefSeq annotation. The RefSeq annotation was downloaded from the UCSC table browser (03/29/2017). Variants annotated by ANNOVAR were filtered for protein-altering events including non-synonymous SNVs, frameshift INDELs, non-frameshift INDELs, and stop-loss. For label-free data, we built a customized database for each individual tumor sample based on germline and somatic variants from matched WES data. For TMT or iTRAQ data, we built a customized database for each TMT or iTRAQ multiplexed sample based on germline and somatic variants from WES data derived from corresponding tumor samples in the TMT or iTRAQ plex. Customized database construction was performed using our JAVA-based tool, customProDBJ. customProDBJ is easier to use compared with our previously published R-based tool customProDB[44]. In addition, customProDBJ includes a redundancy removal feature that is particularly useful for constructing non-redundant customized databases in TMT and iTRAQ studies. For each TMT or iTRAQ sample, customProDBJ takes all ANNOVAR annotated files associated with the tumor samples in the TMT plex or iTRAQ plex as input. The same variant from different tumor samples is identified and only one variant protein sequence for each variant is retained in the customized database for the TMT or iTRAQ sample. The source code of customProDBJ is available at https://github.com/bzhanglab/customprodbj.

**MS/MS searching**. The MS/MS data in mzML format were converted into MGF files using MSconvert (ProteoWizard, version 3.0.19014)[45]. The MS/MS data were searched using three search engines (X! Tandem v2017.2.1.2, MS-GF + v2018.10.15, and Comet v2018.01 rev. 4) against protein databases with decoy protein sequences. For the label-free data set, the following parameters were used for database searching: Fixed modifications, Carbamidomethyl (C); variable modifications, Oxidation (M); Precursor ion mass tolerance, 20 ppm; MS/MS mass tolerance, 0.05 Da; Enzyme specificity, trypsin; maximum missed cleavages, 2. For the TMT data set, parameters for database searching were set as follows: Fixed modifications, Carbamidomethyl (C), TMT 10-plex (K) and TMT 10-plex (N-term); variable modifications, Oxidation (M); Precursor ion mass tolerance, 20 ppm; MS/MS mass tolerance, 0.05 Da; Enzyme specificity, trypsin; maximum missed cleavages, 2. For the iTRAQ data set, parameters for database searching were set as follows: Fixed modifications, Carbamidomethyl (C), iTRAQ 4-plex (K) and iTRAQ 4-plex (N-term); variable modifications, Oxidation (M); Precursor ion mass tolerance, 20 ppm; MS/MS mass tolerance, 0.05 Da; Enzyme specificity, trypsin; maximum missed cleavages, 2. For all data sets, only peptides with length between 7 and 45 were considered.

**FDR estimation**. We used the target-decoy search strategy[17] implemented in PGA[46] (https://github.com/wenbostar/PGA) to estimate FDR. Reversed target protein sequences were used to build a decoy database. FDR estimation was calculated by dividing the number of decoy hits by the number of target hits above a score threshold. All the results were filtered with 1% FDR at peptide level. For peptide level FDR calculation, only the best scoring PSM for each peptide was used. We compared three FDR estimation methods as described below.

For global FDR estimation, the MS/MS data were searched against customized databases which contain both reference and variant protein sequences. PSMs involving both reference and variant peptide sequences were combined together for FDR estimation. The FDR was estimated according to the following equation:

$$FDR = \frac{D}{T} \tag{1}$$

Where $D$ is the number of identified decoy peptides with scores above a score threshold and $T$ is the number of identified target peptides with scores above the same threshold.

For separate FDR estimation, the MS/MS data were searched against customized databases, which contain both reference and variant protein sequences. If an identified peptide could be mapped to the reference protein database, it was defined as a reference peptide. PSMs involving reference and variant peptide sequences were separated into two groups, and FDR estimation was calculated for the two groups separately. The FDR for variant peptides was estimated according to the following equation[18]:

$$FDR_n = \frac{D^+ * \frac{D_n}{D}}{T_n^+} \tag{2}$$

where $D^+$ is the number of identified decoy peptides with scores above a score threshold, $T_n^+$ is the number of identified variant peptides in the target database above the score threshold, $D_n$ is the number of matched decoy variant peptides, and $D$ is the total number of matched decoy peptides. $D_n/D$ is an approximation for the fraction of variant sequences in the search space.

For two-stage FDR estimation, the MS/MS data were searched against the reference protein database in the first stage. The confidently identified spectra with 1% FDR were removed. In the second stage, the remaining spectra were searched against the variant protein sequence databases. The FDR estimation for variant peptides was based on the search results from the second stage. The equation for global FDR estimation was used for both stages.

**PepQuery validation**. The identified variant peptides were further validated using PepQuery (http://pepquery.org/)[23]. For each data set, the database searching parameters described above were used. Hyperscore was used for PSM scoring, unrestricted modification searching-based filtering was enabled, and only PSMs with a $p$ value ≤ 0.01 were considered as true identifications.

**Spectra annotation**. Spectra were annotated using PDV (1.6.0) (http://pdv.zhang-lab.org/)[47] and all the annotated spectra for variant peptides are available at http://pdv.zhang-lab.org/data/download/nc_2019_paper/.

**Deep learning-based retention time prediction**. Using the automatic deep learning and transfer learning techniques, we developed AutoRT, a peptide sequence-based RT prediction tool.

Each peptide was encoded into a matrix using one-hot encoding[48] with a fixed length. Specifically, each amino acid was represented as a binary vector of all zeros except for one entry, which was set to one to indicate the category of the amino acid (one-hot encoding). By default, the fixed length was the length of the longest identified peptide. Amino acid with modification was encoded as a different character so that modified peptide can also be included in the prediction.

A genetic algorithm[49] was developed to automatically search neural architectures for RT prediction. The algorithm was specifically designed to search different convolutional neural networks (CNN)[50] combined with bidirectional gated recurrent unit (GRU) networks[51] for regression. CNNs are variants of deep neural networks (DNNs) that learn high levels of abstractions from multiple layers of nonlinear transformations. CNNs contain convolution and pooling layers that extract sequence features at different spatial scales, and a weight-sharing strategy is used to capture local patterns in protein sequence data. GRU networks are also variants of DNNs that are capable of capturing long-term dependencies between amino acids. A dropout layer with different dropout rates was also considered in the neural architectures to avoid overfitting.

Our genetic algorithm involved constructing an initial generation of individuals (deep neural networks) and performing genetic operations to allow them to evolve in a genetic process. Each DNN structure was represented as a fixed-width genome encoding information about the network's structure. In our setup, a model included a number of convolutional layers, a number of dense layers, an optimizer, and a fixed bidirectional GRU layer with 50 units. The convolutional layers could be evolved to include varying numbers of feature maps, different activation functions, varying proportions of dropout, and whether to perform batch normalization and/or max pooling. The same options were available for the dense layers with the exception of max pooling. The detailed search space is described in

Supplementary Data 11. We further defined several standard genetic operations, i.e., selection, mutation and crossover. The quality of each individual was determined by its MSE on a validation data set. Throughout the genetic process, we evaluated each individual (i.e., network structure) by training it from scratch. The genetic process came to an end after a fixed number of generations. By default, the generation size was 20, and the population size for each generation was 50, and a maximum of 20 epochs and early stop were used for training.

A large public data set[52] (PXD006109) was downloaded from PRIDE[53] and was used for neural architecture search on three NVIDIA Titan XP GPUs. The top 10 best neural architectures were selected based on validation MSE. Ten models were trained based on the top 10 best neural architectures, respectively, with a maximum of 100 epochs and a batch size of 64. The default learning rate was used for each model and early stop was used in the training and the best trained model was saved for each architecture based on validation MSE. The top 10 trained models were taken as the base models for transfer learning. We used the Keras library (v2.2.4, http://keras.io/) along with Tensorflow (v1.13.1, www.tensorflow.org) for implementation.

The top 10 models trained using the large public data set were fine-tuned without freezing any layers to train LC-MS/MS run-specific models using known peptides identified in each run and their RTs with the transfer learning strategy. To ensure high quality of the training data, only known peptides identified by at least two out of the three search engines with 1% FDR at both PSM and peptide levels were included. Known peptides for which the difference between the maximum observed RT and minimum observed RT was >3 min were removed from the training data. All remaining identified known peptides were randomly divided into two parts. The first part contained 90% of the peptides and was used for training. The second part contained 10% of the peptides and was used for independent testing. During the training process, 10% of the training data was randomly selected as validation data. A maximum of 40 epochs was used for training and early stop was used, and the best model was selected based on the validation data for each run. The top 10 models were ensembled using averaging with outliers removed based on the interquartile range (IQR) algorithm. Specifically, the first quantile (Q1), the third quantile (Q3), and the interquartile range (IQR, i.e., Q3–Q1) of the predicted RTs for each peptide were calculated, and predicted RTs outside of the boundaries of Q1–1.5∗IQR and Q3+1.5∗IQR were excluded. Then, the average was calculated from the remaining predicted RTs to represent the final predicted RT for each peptide. The ensemble model was used to predict RT for the variant peptides identified from the same LC-MS/MS run by different methods.

To evaluate the performance of AutoRT, we compared AutoRT with four recently published RT prediction tools, including three deep learning-based tools (Prosit[27], DeepMass[36], and GuanMCP2019[37]) and one traditional machine learning-based tool, GPTime[30]. Three large public data sets were used for evaluation. The first data set was downloaded from PRIDE with accession number PXD006109, which contains 136,791 peptides and was described in the previous section. The second data set was downloaded from SWATHAtlas (http://www.swathatlas.org/) with accession number SAL0031[54] and contains 159,345 peptides. The third data set was also downloaded from SWATHAtlas with accession number SAL0014[55] and contains 124,710 peptides. Each data set was divided into two parts. The first part contained 90% of the peptides and was used for training. During the training process, 10% of the training data was randomly selected as validation data. The second part contained 10% of the peptides and was used for independent testing. All the tools compared in this study were trained from scratch using the same training data used for AutoRT. Because GPTime cannot handle large training data, we randomly selected 10,000 peptides from the first part of each data set as training data for GPTime. We used median absolute error (MAE) according to the following equation to evaluate the performance of RT prediction models.

$$\text{MAE} = \text{median}(|RT_{observed} - RT_{predicted}|) \qquad (3)$$

Where $RT_{observed}$ is the experimental $RT$ for the spectrum that a peptide was identified from, and $RT_{predicted}$ is the RT predicted by deep learning models for the peptide.

For the three CPTAC data sets, we also compared AutoRT with GPTime using the same method.

**Determining observed RT for a peptide for model training.** Each MS/MS (MS2) spectrum or MS2 scan is associated with a distinct RT. Given that MS2 scans can sometimes be taken rather early or rather late in the peak elution, the RT of the MS1 peak maximum of the respective precursor ion can better represent the observed retention time for a peptide. However, the search results from the peptide identification tools used in this study (MS-GF+, X!Tandem, and Comet) do not contain the RT of the MS1 peak maximum of the respective precursor, and it is time consuming to extract MS1 peak maximum-based RT from the raw data for all experiments. We considered an alternative option, which is to use the average of RTs from all spectra identified from the same run with 1% FDR at both PSM and peptide levels for a peptide to represent observed RT for the peptide. To evaluate the accuracy of this MS2-based method, we selected identification results from three runs from each of the three CPTAC data sets and extracted the RT of MS1 peak maximum of the respective precursor ions of identified peptides using FlashLFQ (v1.0.2)[56]. As shown in Supplementary Figs. 8–10, the difference between the RTs estimated based on the MS2 and the MS1 methods was <10 sec for >90, 87, and 95% of the peptides identified in the label-free, TMT, and iTRAQ data, respectively. Moreover, the accuracies of AutoRT were similar when the two types of observed RTs were used for the analysis.

Therefore, the average RT for all spectra identified from the same run for a peptide was used to determine the observed RT for a peptide.

**Retention time prediction for variant peptides.** Theoretical RT for variant peptides were predicted using the ensemble models trained using the transfer learning strategy from the experimental runs where the variant peptides were identified.

**Quality evaluation for variant peptide identifications.** The quality of variant peptide identifications was evaluated using the absolute RT error of the best PSM reported from each search engine for each peptide.

**Neoantigen prediction.** We used Optitype (v1.3.1)[57] to perform HLA genotyping for each sample based on WXS data. Then we used netMHCpan (v4.0)[58] to predict HLA-peptide binding affinity for somatic mutation-derived variant peptides with a length between 8–11 amino acids. The HLA-peptides with IC$_{50}$ binding affinity ≤150 nM were considered to be neoantigens.

**NeoFlow implementation.** The NeoFlow for neoantigen prediction and prioritization includes four modules: (1) variant annotation and customized database construction. Variants in the VCF format are annotated using ANNOVAR, and the annotated files are used as input to customProDBJ in order to build a customized protein database; (2) Variant peptide identification including MS/MS searching, FDR estimation, PepQuery validation, and optional RT-based validation. Three search engines, MS-GF+, X!Tandem, and Comet, are available in NeoFlow for peptide identification. FDR estimation is performed based on the global FDR estimation method. All variant peptides passing the 1% global FDR threshold are further validated using PepQuery, and only those with PepQuery $p$ value ≤0.01 are retained. In addition, if germline variants are considered in the customized database construction, a somatic variant peptide will be removed if it can be exactly mapped to a germline variant peptide. For RT-based validation, an RT prediction model is trained for each single run using all identified reference peptides as training data for AutoRT. The trained model is then used to predict the RTs of the variant peptides. The difference between observed RT and predicted RT is taken as an additional quality metric for each PSM. Because AutoRT requires GPUs that are not available in many proteomics labs, this feature is optional in NeoFlow; (3) HLA typing. Because HLA-peptide binding affinity is affected by both peptide sequence and HLA type, HLA type needs to be specified as an input for binding affinity prediction in step 4. HLA typing in NeoFlow is performed by Optitype, which uses WXS data in the FASTQ format as input; (4) MHC-binding prediction and neoantigen prioritization. netMHCpan[58] is used to predict HLA-peptide binding affinity for all somatic mutation-derived variant peptides with a length between 8–11 amino acids, except for those that can be exactly matched to a reference protein. Following previous publications[15,59], NeoFlow uses IC$_{50}$ binding affinity of 150 nM as a threshold to identify HLA-peptide pairs with strong binding affinity. This information is combined with variant peptide identification information acquired from steps 1–3 for neoantigen prioritization. NeoFlow was implemented into a workflow using nextflow (https://www.nextflow.io/)[60] and is available at https://github.com/bzhanglab/neoflow.

**Application of NeoFlow to immunopeptidomics data.** A published immuno-peptidomics data set[38] was re-analyzed using NeoFlow. The data set includes paired exome sequencing and immunopeptidomics data from five tumors samples. The MS/MS data for the five samples were downloaded from PRIDE under accession number PXD004894. The raw MS/MS data were converted into MGF files using MSconvert (ProteoWizard, version 3.0.19014). We directly used the HLA types reported for the five samples in the original study. The somatic variants of the five samples reported in the original study were extracted from the Supplementary Data 5 of the publication. We used NeoFlow to annotate these somatic variants and build a customized protein database for each sample. The three search engines MS-GF+, X!Tandem, and Comet were used for MS/MS searching. The search parameters were similar to those used in the original publication. The search results were filtered with 1% FDR using the global FDR estimation method. The retained somatic peptides were further validated using PepQuery. In order to speed up PepQuery-based validation of variant peptides in immunopeptidomics data, we updated PepQuery to make use of the matched peptides for each spectra identified by Open-pFind[61]. Specifically, we searched the MS/MS data against the reference protein database using Open-pFind with the open search mode. Then in the steps of competitive filtering based on reference sequences and competitive filtering based on unrestricted post-translational modification searching in PepQuery, we directly used the matched peptides for each spectrum from Open-pFind search as candidates. AutoRT was used to predict RT for identified variant peptides in order to calculate RT errors for additional PSM filtering.

## Data availability

Three large-scale proteomics data sets from a colon cancer study[10] and a breast cancer study[7] were downloaded from the CPTAC data portal, including a colon cancer label-free data set (https://cptac-data-portal.georgetown.edu/cptac/s/S037), a colon cancer TMT data set (https://cptac-data-portal.georgetown.edu/cptac/s/S037), and a breast cancer iTRAQ data set (https://cptac-data-portal.georgetown.edu/cptac/s/S015). Somatic variants and germline variants for colon cancer samples were from the original

publication[10]. Somatic variants and germline variants for breast cancer samples were downloaded from the genomic data commons (GDC, https://portal.gdc.cancer.gov/). The immunopeptidomics data set used in this study was downloaded from PRIDE (https://www.ebi.ac.uk/pride/) with accession number PXD004894. The somatic variants of the five samples included in the immunopeptidomics data set reported in the original study were extracted from the Supplementary Data 5 of the publication. The three large public data sets used for retention time prediction model training and evaluation were downloaded from public databases. The first data set was downloaded from PRIDE with accession number PXD006109. The second data set was downloaded from SWATHAtlas (http://www.swathatlas.org/) with accession number SAL00031[54]. The third data set was also downloaded from SWATHAtlas with accession number SAL00141[55]. The source data underlying Figs. 2b–d, 4–6, and 7a–c are provided as a Source Data file. All other data are available from the corresponding author upon reasonable request.

## Code availability

The source code of NeoFlow is available at https://github.com/bzhanglab/neoflow. AutoRT is available at https://github.com/bzhanglab/AutoRT/.

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

## Acknowledgements

This study was supported by the National Cancer Institute (NCI) CPTAC award U24 CA210954, the Cancer Prevention & Research Institutes of Texas (CPRIT) award RR160027, and funding from the McNair Medical Institute at The Robert and Janice McNair Foundation. We thank Jonathan T. Lei, Sara R. Savage, and Eric Jaehnig for proofreading the manuscript. We gratefully acknowledge the support of NVIDIA Corporation with the donation of the Titan Xp GPU used for this research. B.Z. is a CPRIT Scholar in Cancer Research and a McNair scholar.

## Author contributions

B.Z. and B.W. conceived and designed the study. B.W. designed and implemented AutoRT. B.W., K.L., and Y.Z. implemented NeoFlow. B.W. and K.L. performed data analysis. B.W., K.L., and B.Z. interpreted the data. B.W. and B.Z. wrote the manuscript. All authors read and approved the final manuscript.

## Competing interests

The authors declare no competing interests.
