## [Peer Review File · Nature Communications]

Reviewers' comments:

Reviewer #1 (Remarks to the Author):

The authors present a description of a process for verifying putative identification of variant peptide sequences through integration of genomic and mass spectrometry (MS)-based proteomics data (proteogenomics). Proteogenomics has taken on increased importance in recent years, as an approach to identify expressed variant protein sequences, as well as peptide sequences which may be presented to the immune system as neoantigens, which could represent targets via immunotherapy. The identification of peptide variants is fraught with potential for false positive matches, and as such strict computational measures are needed to attempt to minimize false-positives and provide researchers the most confident results possible for further validation. The authors have taken a sensible route to develop a novel workflow that rigorously identifies putative neoantigens; such a workflow would have significance as a tool for researchers pursuing proteogenomics and neoantigen detection in cancer studies. As part of developing this workflow, the authors provide a new tool called AutoRT, which utilizes the LC retention time of peptides identified by LC-MS/MS analysis, and predicts the correctness of the match of an MS/MS spectrum to a peptide based on a comparison of its measured retention time versus the predicted retention time from AutoRT. They use AutoRT information to validate their workflow for filtering proteogenomic results and ensuring confident results. AutoRT was developed via machine learning on a very large amount of available LC-MS/MS data, and the authors show it has advantages over other existing methods for retention time prediction.

Overall, this is a well written manuscript describing an advanced study addressing a significant research problem. This software has novelty and should see a high amount of usage by the research community, with this publication acting as a foundational description. There are some comments and suggestions for the authors to address before this is published, as detailed below.

Major comments

1. The authors have done nice work in developing AutoRT for predicting retention time of peptides identified via MS/MS and sequence database searching. They show it outperforms another standard program for retention time prediction. A few comments and questions do come up related to AutoRT.

It seems that the primary motivation for developing AutoRT is to confirm the ability of their workflow utilizing PepQuery to identify the most confident variant peptides from proteogenomic studies. Ultimately, however the method for neoantigen peptide identification and prediction does not seem to use AutoRT or RT values for the peptides. It begs the question as to whether AutoRT could be used by others easily in validating matches from MS/MS data for proteogenomic or other proteomic studies:

- Can the algorithm be utilized with smaller amounts of data that is generally generated in most MS-based proteomic studies (not at the level of CPTAC studies?). A more clear statement on the usefulness to the broader MS-proteomics community would be helpful – perhaps it is only useful for very large datasets, which would be good to know. This aspect of the work seems a bit opaque and without much focus in the paper. Is there another publication planned which will further describe AutoRT and its use?
- Could AutoRT be used as a way to assess quality of hits to variant sequences on its own, without need to run the results through another filter such as PepQuery? Related to this question, why not use AutoRT in conjunction with PepQuery for even more rigorous filtering of results?
- Is AutoRT available within a workflow or just as a script within a Github repo?

2. The authors have made their workflow for neoantigen identification and prediction available in

Nextflow, with supporting documentation. This is very good. However, it doesn't look like there is example input data supplied for an interested user to test the workflow? It would be very helpful to make even a small, trimmed down amount of data available – perhaps via a Zenodo link or Github?

Minor comments

1. Retention time prediction for LC-MS based proteomics has been around for quite some time. I would suggest citing some of the earliest studies, which utilized this approach to improve results – e.g. see Petritis et al. *Anal. Chem.* 2003, 75, 1039, and *J Am Soc Mass Spectrom.* 2003 Sep;14(9):980-91.

2. It may be useful to describe in a little more detail how the workflow works for neoantigen prediction, for the general audience. For example, why is HLA typing necessary? This will help the reader understand the necessity for specific inputs and how the different software components work together within the workflow.

Reviewer #2 (Remarks to the Author):

The authors have performed an interesting analysis on the different strategies and methods for assessing variant peptide detection in proteogenomics analysis. It also presents recommendations for improving neoantigen detection.

a) The analysis showing the high variability of neoantigen false discovery numbers produced on the same datasets depending on the search engine and FDR strategy used is enlightening.

b) The suggested use of Global FDR instead of separated FDR to increase sensitivity sounds counterintuitive considering the progress in the field in the last five years and the consensus that FDR was being on the whole underestimated. The manuscript suggests that the key to avoid the pitfall of underestimated false discoveries is to use a post-search validation tool such as PepQuery. This is an interesting direction to pursue.

c) AutoRT appears to be a useful tool to provide an MS-search-independent quality parameter for post-search evaluation of the peptide discovery.

While the paper is interesting, there are some issues that need to be addressed:

1) However, the main text does seem to focus more on PepQuery, which is the subject of a previous publication, than on the newly developed AutoRT. A bit more detail should be given to describe the training of AutoRT and its transfer learning strategy, so that future users can easily and correctly implement AutoRT to their own LCMS datasets. What threshold should be used for delta RT? Given a list of peptides and their spectra how would one decide on the accuracy of RT similarity for each peptide.

It is not shown how using RT as an evaluation metric improves accuracy of neoantigen prediction.

2) In fact, the authors have even failed to name AutoRT in the abstract of the manuscript. This is something that should be corrected.

3) Still in the abstract, there is an overstatement in “3 times more potential treatment opportunities”. Even “3 times more putative neoantigens” is likely an exaggeration given that that assumes the proteogenomics expert to be using a rather weak FDR strategy/search engine combination (for example, as shown in fig 7.a, X!Tandem with two stage FDR). The “3 times more potential treatment opportunities” needs to be toned down to “significant improvement in sensitivity towards the discovery of putative neoantigens” or something similar.

4) There could be room for a minor discussion on the performance of different search engines, although the manuscript does address it in a general way recommending several search engines to be combined to increase sensitivity. But, to me, it seems obvious that X!Tandem is borderline inadequate for a proper proteomics search, even more so for a proteogenomics search. On the other hand, it is of note that MSGF+ is more sensitive, but does take more risks in that it does bring in more false positives, albeit this appears to be fully compensated for by using PepQuery post-search.

5) How well can the ensemble model be generalized for new datasets? This seems to be overkill for new datasets. Has a generalized version of the model been tested? Combining peptides from multiple experiment types for training the deep learning model may handle the experiment-specific parameters internally as a replacement for the transfer-learning step.

In the methods section, it is mentioned that the high-confidence known peptides were used for fine-tuning the model and later the reference peptides were used for performance evaluation. This seems to introduce bias because the test set has already been used in the fine-tuning step.

6) The authors explored the search engine and FDR strategy variables. But, having mentioned three post-search quality control tools, SpectrumAI, SAVcontrol and Pepquery, all three tools could have been employed and explored on the three datasets to assess whether the tools are redundant, or if there would be benefit in actually employing all three, or a combination of two of them together.

7) On page 8, under "Retention time-based quality evaluation" it is described that the observed RT is based on the best scoring PSM of each peptide. This seems to me slightly incorrect. Instead the RT of the MS1 peak maximum of the respective precursor ion should be used, given that MS2 scans can sometimes be taken rather early or rather late in the peak elution. If possible to implement, I would expect this to increase the accuracy of AutoRT.

8) Parts of the text when numbers of peptides are given in rather tedious manner can be eliminated given that all numbers are anyway presented in the mentioned figures (case in point, see page 9, 1st paragraph under "Neoantigen prioritization"). It is more interesting to discuss the implications of those results and just show all the numbers in the figures. Maybe a few numbers that highlight a particular conclusion can be kept in sentence form in the main text body.

9) When stating percentages of FDR estimates, using two decimals is meaningless and unnecessarily makes for a harder read. Please reduce the % numbers to a single decimal, or do without decimals altogether.

10) While the authors have clarified that peptides from germline mutations were not used for neoantigen prediction, it is not specified if they also have discarded peptides that has both germline and somatic mutations from the same patient.

There are also few languages issues such as 'an SAV' should be 'a SAV' and others.

Reviewer #3 (Remarks to the Author):

The manuscript submitted by Bo Wen et al introduces a computational workflow for variant peptide identification that is based on a systematically evaluation and quality control of the results of three search engines using three different FDR-estimation strategies on three different datasets. The resulting workflow produces 3 times more neoantigens candidates than the workflow which performed worst in their study. The overall quality of the manuscript and technical evaluation is of high quality and of high relevance. However, I have some minor and major remarks.

Major remarks:

1) The genetic algorithm for finding a neural architecture for RT predictions appears quite complex and given the relatively simple task of retention time prediction over-engineered. The authors should show that this setup is necessary e.g. by showing that the individual generations of the genetic algorithm significantly improved the MSE and an average sized model without fixed epochs

and early stopping would result in significantly lower MSE. In order to show that their method is a significant advancement ("major contribution of this manuscript") over the current state of the art (deep learning), the authors should compare their method to other recently published deep learning approaches for RT prediction. I would expect to see less of a difference between AutoRT and other deep learning approaches in comparison to classical (feature engineering-dependent) machine learning approaches.

2) Why did the authors not use dropout during learning to avoid over fitting of the RT prediction model? Especially the last transfer learning step, where a fixed number of epochs was used with no early stopping, can lead to high levels of overfitting (see Figure 1e). While the bagging should circumvent this to some degree, the authors should provide an analysis which evaluates the level of overfitting (if present). This is especially the case when LC-MS/MS run-specific models are trained where the number of available training data is low.

3) Using the difference in retention time as an additional feature to separate correct from incorrect matches has been proposed and used for quite some time now and is, in itself, not novel. Additionally, deep learning has been applied with success to RT prediction in prior literature. I question the improvement over the state of the art. AutoRT was used here to evaluate whether the identification which are not retained by PepQuery have an overall higher RT deviation compared to those who are retained and was not used as an additional quality metric in the final pipeline, i.e. as part of the FDR control.

4) The authors argue that this study provides "novel insight and clear guidance on the selection of quality control strategies [for variant peptides identification]". However, I have difficulty seeing this. The final proposed workflow makes use of the combined results of all three evaluated search engines, rather than clear guidance. This will obviously lead to higher numbers (as prior literature has shown that combining search results of multiple search engines increases the number of peptides), especially since the authors chose to compare their results against the "most stringent" FDR setting. In addition, the authors write that further investigation is necessary to check the quality of the identified variant peptides (because of the high absolute RT difference some peptides had even after PepQuery filtering). Given this analysis, it is still unclear what the final FDR of the filtered list of peptides is and how many of the variant peptide which have passed PepQuery might still be wrong.

5) For proper comparison, it seems logical to compare and state the increase in comparison to the best performing existing solution (i.e. MS-GF+-Global FDR + PepQuery; ~15% increase) rather than the worst tested setting.

6) Having established a final workflow for neoantigen prediction, I recommend to show its application by re-analyze a study which directly investigated the immunopeptidome (e.g. ref 34).
Minor remarks:

1) Drop-out can be used to simulate a Bayesian prediction and thus a simpler approach to bagging because it does not rely on multiple models being trained. It would be interesting to see whether the chosen approach has any benefit compared to that.

2) In Figure 7 a-c, it is not obvious why a column can represent the same neoantigen but with differing numbers of mutations. Can the author comment on this?

3) Numbers of variant peptides identified across the different datasets varies a lot. I am somewhat surprised by this as for example the TMT dataset identifies significantly more in comparison to the label free approach. Given that most of the mutations are not shared between patients, I would have expect a dilution effect for patient-specific mutations (1 out of 10 samples contains one), which would reduce the chances of triggering a high quality MS/MS on those. Can the authors comment on this? Does, for example, the higher ratio of b-ions help in confidently identifying site specific ions for variant peptides?

4) What are the proportions of decoys variant peptides being filtered out by PepQuery (Fig 1b,d,f)?

5) What is the distribution predicted binding affinities of the neoantigens? Why did the authors chose 150nM as a filter?

6) Can the authors comment on the usefulness of this approach versus directly measuring the presented (neo)antigens (as in ref 34)?

Re: NCOMMS-19-25550-T “Cancer neoantigen prioritization through sensitive and reliable proteogenomics analysis”

REVISIONS IN RESPONSE TO REVIEWERS’ COMMENTS

We thank the reviewers for the insightful comments and constructive suggestions. We have considered all comments and suggestions and revised the manuscript accordingly. For your convenience, we have also included a version with “track changes” in the submission. Please see below for a point by point response to each of the points made by the reviewers. Page numbers listed below are based on the manuscript version without tracked changes.

Reviewer #1 (Remarks to the Author):

The authors present a description of a process for verifying putative identification of variant peptide sequences through integration of genomic and mass spectrometry (MS)-based proteomics data (proteogenomics). Proteogenomics has taken on increased importance in recent years, as an approach to identify expressed variant protein sequences, as well as peptide sequences which may be presented to the immune system as neoantigens, which could represent targets via immunotherapy. The identification of peptide variants is fraught with potential for false positive matches, and as such strict computational measures are needed to attempt to minimize false-positives and provide researchers the most confident results possible for further validation. The authors have taken a sensible route to develop a novel workflow that rigorously identifies putative neoantigens; such a workflow would have significance as a tool for researchers pursuing proteogenomics and neoantigen detection in cancer studies. As part of developing this workflow, the authors provide a new tool called AutoRT, which utilizes the LC retention time of peptides identified by LC-MS/MS analysis, and predicts the correctness of the match of an MS/MS spectrum to a peptide based on a comparison of its measured retention time versus the predicted retention time from AutoRT. They use AutoRT information to validate their workflow for filtering proteogenomic results and ensuring confident results. AutoRT was developed via machine learning on a very large amount of available LC-MS/MS data, and the authors show it has advantages over other existing methods for retention time prediction.

Overall, this is a well written manuscript describing an advanced study addressing a significant research problem. This software has novelty and should see a high amount of usage by the research community, with this publication acting as a foundational description. There are some comments and suggestions for the authors to address before this is published, as detailed below.

Response: We thank the reviewer for the positive comments.

Major comments

1. The authors have done nice work in developing AutoRT for predicting retention time of peptides identified via MS/MS and sequence database searching. They show it outperforms another standard program for retention time prediction. A few comments and questions do come up related to AutoRT.

It seems that the primary motivation for developing AutoRT is to confirm the ability of their workflow utilizing PepQuery to identify the most confident variant peptides from proteogenomic studies. Ultimately, however the method for neoantigen peptide identification and prediction does not seem to use AutoRT or RT values for the peptides. It begs the question as to whether AutoRT could be used by others easily in validating matches from MS/MS data for proteogenomic or other proteomic studies:

- Can the algorithm be utilized with smaller amounts of data that is generally generated in most MS-based proteomic studies (not at the level of CPTAC studies?). A more clear statement on the usefulness to the broader MS-proteomics community would be helpful – perhaps it is only useful for very large datasets, which would be good to know. This aspect of the work seems a bit opaque and without much focus in the paper. Is there another publication planned which will further describe AutoRT and its use?

Response: AutoRT includes two major steps. First, a large public dataset (e.g., PXD006109 from the Mann group [PMID 29735998] was used in this study) is used for deep neural network architecture search to train base models. This first step requires a large amount of training data, but it does not need to be repeated when applying the resulted based models to a new MS-based proteomics study. In the second step, transfer learning is used to fine-tune the base models trained in the first step using new data from each LC/MS-MS run to build run-specific models. In our study, the number of identified peptides for a single run ranged from 700 to 10,000 (Fig. 2b-d), which is typical for MS-based proteomic studies. The small number of peptides identified in each run is in general not enough to train a peptide sequence-based high-performance deep learning model from scratch, and AutoRT addresses this challenge using the transfer learning strategy. Therefore, AutoRT can be applied to any typical MS-proteomics studies for retention time prediction. We have clarified this in the Discussion section (Page 12).

- Could AutoRT be used as a way to assess quality of hits to variant sequences on its own, without need to run the results through another filter such as PepQuery? Related to this question, why not use AutoRT in conjunction with PepQuery for even more rigorous filtering of results?

Response: We thank the reviewer for these excellent questions. During the revision, we have implemented AutoRT as a standalone Python package that can be used to assess quality of hits to variant sequences on its own. Both the package and a tutorial with an example demonstrating the usage of this function are available at <https://github.com/bzhanglab/AutoRT/>. We also added AutoRT into NeoFlow as an optional step in Module 3 so that it can be used in conjunction with PepQuery for even more rigorous filtering. An example demonstrating this feature is available at <https://github.com/bzhanglab/neoflow>. We note that AutoRT requires GPUs, which are not available in many proteomics labs. Therefore, it is included as an optional module in NeoFlow. These new additions are described in the Method section (Page 20-21).

Rather than simply using delta RT to filter PSMs, we believe a more effective approach to improve peptide identification is to incorporate delta RT as a feature into PSM scoring in combination with other features. We have an on-going project on re-scoring PSMs by integrating delta RT with other features using semi-supervised machine learning, which will be reported in a future manuscript.

- Is AutoRT available within a workflow or just as a script within a Github repo?

Response: As mentioned above, AutoRT is now available as a standalone python package that can be used for RT prediction (<https://github.com/bzhanglab/AutoRT>). We also included AutoRT in NeoFlow as an optional step so that the computed delta RT can be used as an additional quality metric for neoantigen prioritization (<https://github.com/bzhanglab/neoflow>).

2. The authors have made their workflow for neoantigen identification and prediction available in Nextflow, with supporting documentation. This is very good. However, it doesn't look like there is example input data supplied for an interested user to test the workflow? It would be very helpful to make even a small, trimmed down amount of data available – perhaps via a Zenodo link or Github?

Response: Excellent suggestion. We have provided examples on using both NeoFlow and AutoRT, and the example datasets are available at corresponding Github repositories.

Minor comments

1. Retention time prediction for LC-MS based proteomics has been around for quite some time. I would suggest citing some of the earliest studies, which utilized this approach to improve results – e.g. see Petritis et al. Anal. Chem. 2003, 75, 1039, and J Am Soc Mass Spectrom. 2003 Sep;14(9):980-91.

Response: Thanks for the information. We have cited the two studies in the introduction section (references 31 and 32).

2. It may be useful to describe in a little more detail how the workflow works for neoantigen prediction, for the general audience. For example, why is HLA typing necessary? This will help the reader understand the necessity for specific inputs and how the different software components work together within the workflow.

Response: We have described the workflow for neoantigen prediction in more detail, including providing explanation on why HLA typing is necessary, in the Methods section under a new subsection “NeoFlow implementation” (Page 20-21).

Reviewer #2 (Remarks to the Author):

The authors have performed an interesting analysis on the different strategies and methods for assessing variant peptide detection in proteogenomics analysis. It also presents recommendations for improving neoantigen detection.

- a) The analysis showing the high variability of neoantigen false discovery numbers produced on the same datasets depending on the search engine and FDR strategy used is enlightening.
- b) The suggested use of Global FDR instead of separated FDR to increase sensitivity sounds

counterintuitive considering the progress in the field in the last five years and the consensus that FDR was being on the whole underestimated. The manuscript suggests that the key to avoid the pitfall of underestimated false discoveries is to use a post-search validation tool such as PepQuery. This is an interesting direction to pursue.

c) AutoRT appears to be a useful tool to provide an MS-search-independent quality parameter for post-search evaluation of the peptide discovery.

Response: We thank the reviewer for clearly summarizing the most important points of the study.

While the paper is interesting, there are some issues that need to be addressed:

1) However, the main text does seem to focus more on PepQuery, which is the subject of a previous publication, than on the newly developed AutoRT. A bit more detail should be given to describe the training of AutoRT and its transfer learning strategy, so that future users can easily and correctly implement AutoRT to their own LCMS datasets. What threshold should be used for delta RT? Given a list of peptides and their spectra how would one decide on the accuracy of RT similarity for each peptide. It is not shown how using RT as an evaluation metric improves accuracy of neoantigen prediction.

Response: We thank the reviewer for this suggestion. In the revised manuscript, we have described in more detail the training of AutoRT and the transfer learning strategy on Page 18-19. We have also implemented AutoRT as a standalone Python package so that it can be easily used for RT prediction in other LC-MS/MS studies.

The primary purpose of using RT prediction in the current study is to provide an independent evaluation metric (delta RT) for evaluating existing quality control methods for variant peptide identification. For this purpose, only relative comparison was required, and there was no need for choosing a delta RT threshold. Based on the relative comparisons, we found that global FDR control followed by PepQuery validation offered the highest sensitivity without compromising the quality of variant peptide identifications.

To directly use RT to improve accuracy of neoantigen prediction, we added an optional AutoRT step into NeoFlow during the revision so that delta RT can be used as an additional filter if GPU is available to run AutoRT. Although this is helpful as shown in our newly added analysis of an immunopeptidome dataset, we agree with the reviewer that it is difficult to choose a fixed threshold for delta RT. Rather than simply using delta RT to filter PSMs, we believe a more effective approach to improve peptide identification is to incorporate delta RT as a feature into PSM scoring in combination with other features. We noted this in the Discussion section (Page 13). We have an on-going project on re-scoring PSMs by integrating delta RT with other features using semi-supervised machine learning, which will need more time to develop and will be reported in a future manuscript.

2) In fact, the authors have even failed to name AutoRT in the abstract of the manuscript. This is something that should be corrected.

Response: Corrected as suggested.

3) Still in the abstract, there is an overstatement in “3 times more potential treatment opportunities”. Even “3 times more putative neoantigens” is likely an exaggeration given that that assumes the proteogenomics expert to be using a rather weak FDR strategy/search engine combination (for example, as shown in fig 7.a, X!Tandem with two stage FDR). The “3 times more potential treatment opportunities” needs to be toned down to “significant improvement in sensitivity towards the discovery of putative neoantigens” or something similar.

Response: Agreed and we have revised the text accordingly.

4) There could be room for a minor discussion on the performance of different search engines, although the manuscript does address it in a general way recommending several search engines to be combined to increase sensitivity. But, to me, it seems obvious that X!Tandem is borderline inadequate for a proper proteomics search, even more so for a proteogenomics search. On the other hand, it is of note that MSGF+ is more sensitive, but does take more risks in that it does bring in more false positives, albeit this appears to be fully compensated for by using PepQuery post-search.

Response: Good suggestion. We have included a new paragraph in the Discussion section on this topic (Page 13): “Although the primary goal of the study was to compare different quality control strategies, our results also revealed the performance difference of different search engines in proteogenomics search. Among the three search engines investigated, MS-GF+ showed the highest sensitivity in variant peptide identification both before and after PepQuery validation (**Fig. 3**). Meanwhile, MS-GF+ also identified higher percentages of variant peptides that failed PepQuery validation, suggesting higher risk in bringing in more false positives when used without PepQuery validation. X!Tandem showed the lowest sensitivity among the three, and we would not recommend using this search engine by itself in proteogenomics search. However, when used together with other search engines, it may still add unique variant identifications to improve the overall sensitivity (**Fig. 7**).”

5) How well can the ensemble model be generalized for new datasets? This seems to be overkill for new datasets. Has a generalized version of the model been tested? Combining peptides from multiple experiment types for training the deep learning model may handle the experiment-specific parameters internally as a replacement for the transfer-learning step.

Response: It is true that combining data from multiple experiment types for training may be able to create a single model that can handle experiment-specific parameters; however, this cannot replace transfer-learning because nonlinear RT shift between different runs occurs frequently even when the same LC system is used for all runs in a study (PMID: 27701844). As shown in Supplementary Fig. S6, X axis represents the observed peptide RTs of fraction 1 of an experiment (#1) from the iTRAQ study and Y axis represents the RTs of the same set of peptides from fraction 1 of another experiment (#30). This figure clearly shows a nonlinear retention time shift, even when the same LC system was used for all experiments in this study. Therefore, a general model

is unlikely to produce good performance for all experimental runs. We have included this information in the Discussion section (Page 12).

In the methods section, it is mentioned that the high-confidence known peptides were used for fine-tuning the model and later the reference peptides were used for performance evaluation. This seems to introduce bias because the test set has already been used in the fine-tuning step.

Response: We are sorry about the confusion. The training of AutoRT consists of two steps. In the first step, 10 base models are trained using a large public dataset (PXD006109). These models are not directly used for RT prediction for new datasets (or experiments) and are served as base models for transfer learning. In the second step, data from a new experiment is used to fine tune the 10 base models using the transfer learning strategy and then the 10 fine-tuned models are ensembled as the final model for experiment-specific RT prediction. In the second step, only a small number of peptides are required to train an ensemble model. In this study, during the transfer learning step, the high-confidence known (reference) peptides were divided into two parts, one part was used for training and the other part was used for independent testing. None of the peptides in testing data was present in training data. Therefore, there was no information leaking during transfer learning. We have added more details in the Methods section to clarify the transfer learning process (Page 18-19).

6) The authors explored the search engine and FDR strategy variables. But, having mentioned three post-search quality control tools, SpectrumAI, SAVcontrol and Pepquery, all three tools could have been employed and explored on the three datasets to assess whether the tools are redundant, or if there would be benefit in actually employing all three, or a combination of two of them together.

Response: To address this comment, we tried to compare PepQuery with SpectrumAI and SAVcontrol during the revision. We contacted the developer of SAVcontrol and found that it was developed in MATLAB and is only compatible with Windows. In addition, it accepts only a specific version of pepXML or Mascot DAT file as input. Because the three search engines (MS-GF+, X!Tandem and Comet) we used in this study cannot generate SAVcontrol-compatible data formats, we were not able to compare PepQuery with SAVcontrol.

Applying SpectrumAI tools to all variant peptides passing global FDR control in the iTRAQ dataset, in which the largest number of variant peptides were identified, validated 29,618, 31,654, and 29,677 variant peptides for the search results from Comet, MS-GF+, and X!Tandem, respectively. These numbers were 6%, 4% and 7% lower than those validated by PepQuery, with 93%, 91%, and 93% overlap. As shown in **Supplementary Fig. S4**, variant peptides uniquely validated by PepQuery (P1) showed similar quality compared with the ones validated by both PepQuery and SpectrumAI (P2); however, those uniquely validated by SpectrumAI (P3) showed obviously higher RT errors. These results suggest that SpectrumAI will unlikely provide significant added value beyond PepQuery for variant peptide validation. We included these results in the revised manuscript (Page 9-10).

7) On page 8, under “Retention time-based quality evaluation” it is described that the observed RT is based on the best scoring PSM of each peptide. This seems to me slightly incorrect.

Instead the RT of the MS1 peak maximum of the respective precursor ion should be used, given that MS2 scans can sometimes be taken rather early or rather late in the peak elution. If possible to implement, I would expect this to increase the accuracy of AutoRT.

Response: We agree with the reviewer that MS2 scans can sometimes be taken rather early or rather late in the peak elution, and using the RT based on the best scoring PSM may be misleading. Unfortunately, the search results from the peptide identification tools used in this study (MS-GF+, X!Tandem, and Comet) do not contain the RT of the MS1 peak maximum of the respective precursor, and it is time consuming to extract MS1 peak maximum-based RT from the raw data for all experiments. During the revision, we considered an alternative option, which is to use the average of RTs from all spectra identified from the same run with 1% FDR at both PSM and peptide levels for a peptide to represent observed RT for the peptide. To evaluate the accuracy of this MS2-based method, we selected identification results from three runs from each of the three CPTAC datasets and extracted the RT of MS1 peak maximum of the respective precursor ions of identified peptides using FlashLFQ (v1.0.2). As shown in **Supplementary Fig. S7-S9**, the difference between the RTs estimated based on the MS2 and the MS1 methods was less than 10 seconds for more than 90%, 87%, and 95% of the peptides identified in the label-free, TMT, and iTRAQ data, respectively. Moreover, the accuracies of AutoRT were similar when the two types of observed RTs were used for the analysis. Therefore, the average RT for all spectra identified from the same run for a peptide was used to determine the observed RT for a peptide. We have included these results in the Methods section under a new subsection “Determining observed retention time for a peptide for model training” (Page 19-20).

8) Parts of the text when numbers of peptides are given in rather tedious manner can be eliminated given that all numbers are anyway presented in the mentioned figures (case in point, see page 9, 1st paragraph under “Neoantigen prioritization”). It is more interesting to discuss the implications of those results and just show all the numbers in the figures. Maybe a few numbers that highlight a particular conclusion can be kept in sentence form in the main text body.

Response: Agreed and we have updated the text accordingly.

9) When stating percentages of FDR estimates, using two decimals is meaningless and unnecessarily makes for a harder read. Please reduce the % numbers to a single decimal, or do without decimals altogether.

Response: Agreed and we have updated the text and Figure 3 accordingly.

10) While the authors have clarified that peptides from germline mutations were not used for neoantigen prediction, it is not specified if they also have discarded peptides that has both germline and somatic mutations from the same patient.

Response: If germline variants are considered in the customized database construction, a somatic variant peptide will be removed if it can be exactly mapped to a germline variant peptide. We have clarified this in the revised manuscript (Page 21).

There are also few language issues such as ‘an SAV’ should be ‘a SAV’ and others.

Response: We went through the manuscript carefully and corrected all language issues we can identify.

Reviewer #3 (Remarks to the Author):

The manuscript submitted by Bo Wen et al introduces a computational workflow for variant peptide identification that is based on a systematically evaluation and quality control of the results of three search engines using three different FDR-estimation strategies on three different datasets. The resulting workflow produces 3 times more neoantigens candidates than the workflow which performed worst in their study. The overall quality of the manuscript and technical evaluation is of high quality and of high relevance. However, I have some minor and major remarks.

Response: We thank the reviewer for the positive comment.

Major remarks:

1) The genetic algorithm for finding a neural architecture for RT predictions appears quite complex and given the relatively simple task of retention time prediction over-engineered. The authors should show that this setup is necessary e.g. by showing that the individual generations of the genetic algorithm significantly improved the MSE and an average sized model without fixed epochs and early stopping would result in significantly lower MSE. In order to show that their method is a significant advancement (“major contribution of this manuscript”) over the current state of the art (deep learning), the authors should compare their method to other recently published deep learning approaches for RT prediction. I would expect to see less of a difference between AutoRT and other deep learning approaches in comparison to classical (feature engineering-dependent) machine learning approaches.

Response: In the neural architecture search using genetic algorithm, 1000 models were generated with a total of 20 generations and 50 models in each generation. As shown in the figure to the right, the quality of the models as measured by validation loss (MSE) improved throughout the generations. The models from the last generation obviously outperformed those randomly generated in the first generation.

During the revision, we compared AutoRT with four RT prediction tools, including three recently published deep learning based RT prediction tools (Prosit [PMID: 31133760], GuanMCP2019 [PMID: 31249099] and DeepMass [PMID: 31133761]) and a classical (feature engineering-dependent) machine tool GPTIME using three large datasets. As shown in **Figure 2A** and **Supplementary Figure S1**, AutoRT outperformed all other three deep learning tools for RT prediction, and all deep learning tools outperformed the classical machine learning approach GPTIME to a large extent. Moreover, within AutoRT, the ensemble models outperformed individual models in all the three datasets, with an average of 24%, 18% and 18% improvement,

respectively (**Supplementary Fig. S2**). This result clearly indicates that combining multiple models can improve the performance of RT prediction. We have included these results in the revised manuscript (Page 6-7).

2) Why did the authors not use dropout during learning to avoid over fitting of the RT prediction model? Especially the last transfer learning step, where a fixed number of epochs was used with no early stopping, can lead to high levels of overfitting (see Figure 1e). While the bagging should circumvent this to some degree, the authors should provide an analysis which evaluates the level of overfitting (if present). This is especially the case when LC-MS/MS run-specific models are trained where the number of available training data is low.

Response: We are sorry this was not clearly described in the original manuscript. We actually used dropout during training. We considered different dropout rates during the neural architecture search as described in **Supplementary Table S11**. In the revised manuscript, we added a sentence to clarify this in the Methods section (Page 17). During the revision, we further added early stopping and described this in the Methods section (Page 18). To evaluate the level of overfitting, we selected three runs from each of the three datasets and compared the prediction errors on the training and independent testing data. The prediction error distributions were comparable between the training and testing data, and only a slight increase of the median was observed in the testing data (**Supplementary Fig. S3**). Therefore, overfitting is not a major issue here. We included this information in the revised manuscript (Page 7).

3) Using the difference in retention time as an additional feature to separate correct from incorrect matches has been proposed and used for quite some time now and is, in itself, not novel. Additionally, deep learning has been applied with success to RT prediction in prior literature. I question the improvement over the state of the art.

Response: As mentioned in our response to comment #1 from the reviewer, we have compared AutoRT with three recently published deep learning-based RT prediction tools and have included the results in the revision.

AutoRT was used here to evaluate whether the identification which are not retained by PepQuery have an overall higher RT deviation compared to those who are retained and was not used as an additional quality metric in the final pipeline, i.e. as part of the FDR control.

Response: We thank the reviewer for this useful comment. During the revision, we added an optional AutoRT step in NeoFlow so that delta RT can be used as an additional filter if GPU is available to run AutoRT. Although this is helpful as shown in our newly added analysis of an immunopeptidome dataset, we believe a more effective approach to improve peptide identification is to incorporate delta RT as a feature into PSM scoring in combination with other features rather than simply using delta RT as an additional filter for PSMs. We noted this in the Discussion section (Page 13). We have an on-going project on re-scoring PSMs by integrating delta RT with other features using semi-supervised machine learning, which will need more time to develop and will be reported in a future manuscript.

4) The authors argue that this study provides “novel insight and clear guidance on the selection

of quality control strategies [for variant peptides identification]”. However, I have difficulty seeing this. The final proposed workflow makes use of the combined results of all three evaluated search engines, rather than clear guidance. This will obviously lead to higher numbers (as prior literature has shown that combining search results of multiple search engines increases the number of peptides), especially since the authors chose to compare their results against the “most stringent” FDR setting. In addition, the authors write that further investigation is necessary to check the quality of the identified variant peptides (because of the high absolute RT difference some peptides had even after PepQuery filtering). Given this analysis, it is still unclear what the final FDR of the filtered list of peptides is and how many of the variant peptide which have passed PepQuery might still be wrong.

Response: We agree with the reviewer that the true FDR of the filtered list of peptides is difficult to know exactly without a controlled experiment. However, we believe our study has provided practical guidance on the selection of quality control strategies. As mentioned in the first paragraph in the Discussion section, “among all quality control strategies investigated, global FDR control followed by PepQuery validation offered the highest sensitivity while identifying high quality variant peptides. We thus recommend this quality control strategy for variant peptide identification and neoantigen prioritization in future proteogenomic studies”.

Although the primary goal of the study was to compare different quality control strategies, our results also revealed the performance difference of different search engines in proteogenomics search. Among the three search engines, MS-GF+ showed the highest sensitivity in variant peptide identification both before and after PepQuery validation (**Fig. 3**). Meanwhile, MS-GF+ also identified higher percentages of variant peptides that failed PepQuery validation, suggesting higher risk in bringing in more false positives when used without PepQuery validation. X! Tandem showed the lowest sensitivity among the three, and we would not recommend using this search engine by itself in proteogenomics search. However, when used together with other search engines, it may still add unique variant identifications to improve the overall sensitivity (**Fig. 7**). We included this information in the Discussion section of the revised manuscript (Page 13). In addition, as mentioned above, we added an optional AutoRT step in NeoFlow during the revision so that delta RT can be used as an additional filter if GPU is available to run AutoRT.

5) For proper comparison, it seems logical to compare and state the increase in comparison to the best performing existing solution (i.e. MS-GF+-Global FDR + PepQuery; ~15% increase) rather than the worst tested setting.

Response: Comparing with the worst tested setting highlights the value of the optimized approach and the problem when suboptimal methods were used for database search and FDR control, whereas comparing with the best tested single search engine setting reveals specifically the added value of combining results from multiple search engines. Both comparisons provide useful information and thus we updated this sentence in the revision: “These numbers represent an average increase of 11% from those reported by MS-GF with global FDR control, the best single search engine setting tested, and an average increase of 151% from those reported by X!Tandem with two-stage FDR control, the worst tested setting.” (Page 10-11)

6) Having established a final workflow for neoantigen prediction, I recommend to show its application by re-analyze a study which directly investigated the immunopeptidome (e.g. ref 34).

Response: Excellent suggestion. We applied the workflow to a published immunopeptidomics study (Ref 35) and included the results in the revised manuscript (Page 11). The RT prediction models trained for this dataset showed similar performance to those trained on the CPTAC datasets (**Supplementary Fig. S5**). NeoFlow identified nine out of the 11 somatic variant peptides reported in the original study and four additional somatic variant peptides (**Supplementary Table 10**). The two somatic variant peptides reported in the original paper but not identified by NeoFlow showed obviously higher absolute RT errors, suggesting the possibility of false positives. Among the four newly identified somatic variant peptides, two have been reported in a recently published reanalysis of the same dataset (Ref 36). These results demonstrate the sensitivity and specificity of NeoFlow in analyzing immunopeptidomics data and the value of RT-based validation as an additional filter to reduce false positives.

Minor remarks:

1) Drop-out can be used to simulate a Bayesian prediction and thus a simpler approach to bagging because it does not rely on multiple models being trained. It would be interesting to see whether the chosen approach has any benefit compared to that.

Response: As mentioned in our response to Comment #2, we already used dropout during training. Moreover, as mentioned in our response to Comment #1, the ensemble models outperformed individual models in all the three datasets, with an average of 24%, 18% and 18% improvement, respectively (Supplementary Fig. S2). This result clearly indicates that combining multiple models can improve the performance of RT prediction. We have included these comparison results in the revised manuscript (Page 6-7).

2) In Figure 7 a-c, it is not obvious why a column can represent the same neoantigen but with differing numbers of mutations. Can the author comment on this?

Response: Sorry about the confusion. In Figure 7 a-c, each column represents a sample, not a neoantigen. Each cell in the figure represents the number of somatic mutations with predicted neoantigens in the corresponding sample. We have clarified this in the legend of Figure 7 in the revised manuscript.

3) Numbers of variant peptides identified across the different datasets varies a lot. I am somewhat surprised by this as for example the TMT dataset identifies significantly more in comparison to the label free approach. Given that most of the mutations are not shared between patients, I would have expect a dilution effect for patient-specific mutations (1 out of 10 samples contains one), which would reduce the chances of triggering a high quality MS/MS on those. Can the authors comment on this? Does, for example, the higher ratio of b-ions help in confidently identifying site specific ions for variant peptides?

Response: There are two major reasons for the highly different variant peptide numbers across the three datasets. First, each sample in the label free, TMT, and iTRAQ data had 6, 12, and 24 fractions, respectively. Second, each iTRAQ and TMT samples include multiplexed tumor samples, leading to higher diversity and increased numbers of variants. The detailed total numbers of identified peptides for both reference peptides and variant peptides are shown in the table below for the label-free and TMT studies, which were generated from the same set of colon cancer samples. Consistent with the reviewer's expectation, label-free data had a slightly higher detection ratio of variant peptides compared with TMT data as shown below.

# of reference peptides	# of variant peptides	Dataset	FDR method	Software	Detection ratio of variant peptides
90979	1389	Label free	Global FDR	Comet	1.53%
128797	1255	TMT	Global FDR	Comet	0.97%
107617	1617	Label free	Global FDR	MS-GF+	1.50%
142013	1497	TMT	Global FDR	MS-GF+	1.05%
84958	1377	Label free	Global FDR	X!Tandem	1.62%
124652	1285	TMT	Global FDR	X!Tandem	1.03%

4) What are the proportions of decoys variant peptides being filtered out by PepQuery (Fig 1b,d,f)?

Response: To address this comment, we examined the search results from MS-GF+ with global FDR estimation on the iTRAQ dataset because this analysis reported the largest number of variant peptide identifications. Across all samples, an average of 86% decoy variant peptides were filtered out by PepQuery.

5) What is the distribution predicted binding affinities of the neoantigens? Why did the authors chose 150nM as a filter?

Response: The predicted binding affinities of all neoantigens without filtering are shown in the figure below for each dataset, respectively. The binding affinity threshold of 150 nM was chosen because it has been used in many previous publications (PMID: 24891321, PMID: 28484631, PMID: 30568305, PMID: 28678778, PMID: 27187383, PMID: 31031003 and PMID: 31675502) and is considered as an indication of a strong binder. This has been clarified in the Method section (Page 21).

6) Can the authors comment on the usefulness of this approach versus directly measuring the presented (neo)antigens (as in ref 34)?

Response: NeoFlow can be applied to both global proteomics data from tumor tissues and immunopeptidomics data (directly measuring the presented (neo)antigens). Analysis of both types of data requires HLA typing, MS/MS searching, and binding affinity prediction. Immunopeptidomics data provide direct evidence of both expression and presentation of somatic variant peptides, and when immunopeptidomics data is not available, global proteomics data from tumor tissues can provide expression evidence for somatic variants. We included this information in the Discussion section (Page 13).

REVIEWERS' COMMENTS:

Reviewer #1 (Remarks to the Author):

The authors have addressed my concerns well.

My only suggestion for revisions would be to supply the delta RT values for the predicted neoantigen peptides described on the bottom of page 10 from the label free, iTRAQ and TMT datasets. Do all of these predicted neoantigens also have satisfactory delta RT values, which would supply even more confidence to these peptides.

Reviewer #3 (Remarks to the Author):

The revised manuscript submitted by Bo Wen et al has been substantially improved. I am particularly delighted to see that the authors have toned down the wording with respect to the 3-times improvement they observed. The authors have largely addressed my previously raised major and minor concerns.

Open issues are:

The authors state that "Although RT has been suggested as a useful constraint in database searching (32), it is typically not used in peptide identification and is independent of the FDR estimation." (2003). While this is in principle correct that RT is not commonly used, since 2003 (ref 32) additional research was done on integrating RT information into the proteomics workflow and the authors are not the first to try/do this in general. I urge the authors to cite the relevant literature here as well and rephrase the sentence to account for that.

Adding AutoRT solely as an optional feature in NeoFlow begs the question of what the contribution of RT information actually is in the proposed workflow. How much does AutoRT add in comparison to using the previous version of PepQuery. Does the here proposed workflow significantly improve the status quo.

One additional minor issue which came up with these revisions, it is not entirely clear to me whether the other deep learning models were re-trained individually on the three tested datasets or pre-trained models were used. In case pre-trained models were used, the comparison is flawed by their own argumentation and a comparison between transfer learned and non-transfer learned models obviously favors the first. This should then be mentioned in the manuscript.

Re: NCOMMS-19-25550A “Cancer neoantigen prioritization through sensitive and reliable proteogenomics analysis”

REVISIONS IN RESPONSE TO REVIEWERS’ COMMENTS

We thank the reviewers for the comments and suggestions. We have considered all comments and suggestions and revised the manuscript accordingly. Please see below for a point by point response to each of the points made by the reviewers.

Reviewer #1 (Remarks to the Author):

The authors have addressed my concerns well.

My only suggestion for revisions would be to supply the delta RT values for the predicted neoantigen peptides described on the bottom of page 10 from the label free, iTRAQ and TMT datasets. Do all of these predicted neoantigens also have satisfactory delta RT values, which would supply even more confidence to these peptides.

Response: Thanks for the useful suggestion. In the revised manuscript, we have added the delta RT values for predicted neoantigen peptides from all the three proteomics datasets in Supplementary Data 7, 8 and 9. We also summarized the data in the newly added Supplementary Figure 5, which shows that the median absolute RT errors for these neoantigen peptides are comparable with those for reference peptides (Figure 2b-d), suggesting an overall high quality of the neoantigen identifications.

Reviewer #3 (Remarks to the Author):

The revised manuscript submitted by Bo Wen et al has been substantially improved. I am particularly delighted to see that the authors have toned down the wording with respect to the 3-times improvement they observed. The authors have largely addressed my previously raised major and minor concerns.

Open issues are:

The authors state that “Although RT has been suggested as a useful constraint in database searching (32), it is typically not used in peptide identification and is independent of the FDR estimation.” 2003). While this is in principle correct that RT is not commonly used, since 2003 (ref 32) additional research was done on integrating RT information into the proteomics workflow and the authors are not the first to try/do this in general. I urge the authors to cite the relevant literature here as well and rephrase the sentence to account for that.

Response: We thank the reviewer for this suggestion. We did an extensive literature review and added all relevant papers (PMID: 29863353, 15359729, 31260443 and 17622186) in the introduction section on page 5. “*Although a few studies have showed the value of integrating RT information into the proteomics data analysis workflow^{26, 32-35}, RT is typically not used in peptide identification*”.

Adding AutoRT solely as an optional feature in NeoFlow begs the question of what the contribution of RT information actually is in the proposed workflow. How much does AutoRT add in comparison to using the previous version of PepQuery. Does the here proposed workflow significantly improve the status quo.

Response: To evaluate the added value of AutoRT in the proposed workflow, we added the delta RT values for predicted neoantigen peptides from all the three proteomics datasets in Supplementary Data 7, 8 and 9. We summarized the data in the newly added Supplementary Figure 5 and added a new paragraph in the results section on Page 11:

“Across all three datasets, putative neoantigens identified by different search engines using different FDR estimation methods in combination with PepQuery filtering showed an average median absolute RT error of 0.64 minutes (Supplementary Fig. 5, Supplementary Data 7-9), which was comparable to those for reference peptides (Fig. 2b-d). Despite the overall high quality of these putative neoantigen identifications, there were some clear outliers (Supplementary Fig. 5). On average, 7% of these identifications showed an RT error higher than 5 minutes and may thus require more critical evaluation.”

We further clarified this result in the discussion section on Page 13:

“The RT errors of variant peptides that passed PepQuery validation were comparable to those of reference peptides in corresponding datasets; however, some variant peptides had high RT errors. This may be explained by different reasons such as inaccurate RT prediction and wide elution time range for some peptides, and false variant peptide identification is also a possible explanation. Therefore, the RT errors included in the final report of NeoFlow provide orthogonal information that facilitates candidate prioritization for experimental validation.”

AutoRT was added only as an optional feature in NeoFlow because it requires GPUs that are not available in many proteomics labs. Although these labs will not be able to compute delta RTs for their own datasets, they can still use our standardized pipeline optimized in this study using an AutoRT-derived evaluation metric. To fully incorporate RT in peptide identification, a more effective approach is to add delta RT as a feature in a scoring algorithm so that it can be combined with other factors for PSM scoring. To fully evaluate the added benefit, we will need to identify a new evaluation metric that is independent of PSM scoring and RT, which could be an interesting topic for a future study. We clarified this in the discussion section on Page 13:

“Rather than using RT errors as an optional feature to filter PSMs, a more effective approach to improve peptide identification is to incorporate AutoRT-derived delta RT as a feature into PSM scoring in combination with other features. However, such implementation will require graphics processing units (GPUs) for PSM scoring, and thus will be more useful when GPUs are widely accessible in proteomics laboratories.”

One additional minor issue which came up with these revisions, it is not entirely clear to me whether the other deep learning models were re-trained individually on the three tested datasets

or pre-trained models were used. In case pre-trained models were used, the comparison is flawed by their own argumentation and a comparison between transfer learned and non-transfer learned models obviously favors the first. This should then be mentioned in the manuscript.

Response: We are sorry this was not clear in the previous version of the manuscript. In the revised manuscript, we have clearly pointed out that “*All the tools compared in this study were trained from scratch using the same training data used for AutoRT.*” (page 19).